# *NeoWorld*: Neural Simulation of Explorable Virtual Worlds via Progressive 3D Unfolding

## Abstract

We introduce NeoWorld, a deep learning framework for generating interactive 3D virtual worlds from a single input image. Inspired by the *on-demand worldbuilding* concept in the science fiction novel *Simulacron-3 (1964)*, our system constructs expansive environments where only the regions actively explored by the user are rendered with high visual realism through object-centric 3D representations. Unlike previous approaches that rely on global world generation or 2D hallucination, NeoWorld models key foreground objects in full 3D, while synthesizing backgrounds and non-interacted regions in 2D to ensure efficiency. This hybrid scene structure, implemented with cutting-edge representation learning and object-to-3D techniques, enables flexible viewpoint manipulation and physically plausible scene animation, allowing users to control object appearance and dynamics using natural language commands. As users interact with the environment, the virtual world progressively unfolds with increasing 3D detail, delivering a dynamic, immersive, and visually coherent exploration experience. NeoWorld significantly outperforms existing 2D and depth-layered 2.5D methods on the WorldScore benchmark.

## 1 Introduction

In the 1964 science fiction novel *Simulacron-3*, the protagonist, Douglas Hall, navigates a virtual simulation of 1937 Los Angeles, where he discovers that only the areas he actively interacts with are rendered in detail. This *on-demand worldbuilding* concept inspires our **NeoWorld** framework, which leverages neural networks to construct an infinite, interactive virtual world from a single image. In NeoWorld, the simulated environment is initially represented in 2D and progressively evolves into detailed 3D models as users engage with it. This user-driven rendering strategy provides immersive experiences while maintaining computational efficiency.

NeoWorld builds upon recent progress in learning-based interactive world generation (Yu et al., 2025; 2024), which has demonstrated promising capabilities in open-vocabulary and view-consistent environment synthesis. These approaches, though effective for infinite static rendering or camera-path navigation, are not designed for interactive exploration where users may dynamically uncover or manipulate different parts of the world. They often rely on 2D extrapolation (Rombach et al., 2022; Zhuang et al., 2024; Corneanu et al., 2024) or 2.5D layered representations (Yu et al., 2025), which result in noticeable artifacts under large viewpoint changes and fall short in supporting dynamic, interactive scene manipulation.

*How can we enable AI systems to simulate infinitely expandable digital worlds with both high-fidelity visual realism and physically grounded dynamics?* This requires meeting two key conditions. First, the scene should be object-centric, allowing fine-grained manipulation and interaction with individual entities. Second, the system must balance 3D immersion with computational efficiency. While full 3D modeling (Qiu et al., 2024; Xie et al., 2024; Guan et al., 2022) supports physics-consistent interaction and coherent view synthesis, it is often computationally expensive. To address this, NeoWorld introduces a hybrid object-centric scene structure that progressively unfolds 2D object representations into 3D, guided by object proximity along the camera trajectory or user-specified prompts.

Unlike prior approaches (Yu et al., 2025; 2024), we propose a deep learning framework that begins with an inverse rendering pipeline, reconstructing the input image using lightweight, object-centric 2D representations enriched with instance-level semantic information. As shown in Fig. 1, this design enables precise object selection in response to novel scene descriptions specified by the user. To

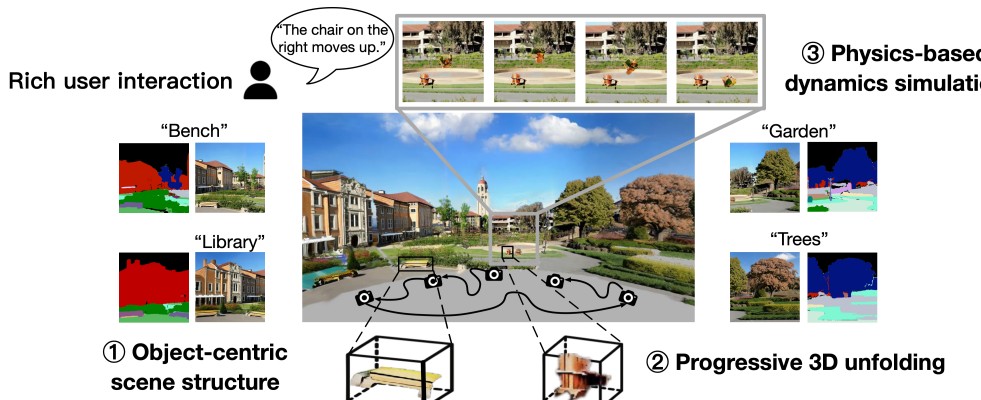

Figure 1: **An overview of our approach.** NeoWorld constructs an infinitely expandable virtual world by integrating object-centric representation learning, image-to-3D reconstruction, and dynamics simulation. It progressively unfolds a 3D scene through user exploration or natural language commands

enhance physical realism and facilitate user interaction within the constructed digital environment, such as changing viewpoints or controlling object motions via natural language, we first incorporate large language models (LLMs) (Team et al., 2023; Bai et al., 2023; Achiam et al., 2023; Liu et al., 2024a) for on-demand object selection, and then apply an image-to-3D technique (Wu et al., 2025) to progressively convert frequently accessed or viewpoint-relevant objects into full 3D representations. These 3D representations are then tightly aligned with the original 2D image at the object level, enabling seamless integration and consistent visual coherence.

NeoWorld outperforms prior 2D (Hong et al., 2023; Wan et al., 2025) and 2.5D (Yu et al., 2025; 2024) methods in interactive world generation, delivering more consistent 3D rendering quality and greater user engagement. In summary, the main contributions of NeoWorld are as follows:

- NeoWorld is a pilot study on *interactive world generation with 3D dynamics* from a single image. Its core idea is to enhance virtual realism while preserving computational efficiency by **progressively unfolding 3D content** along user exploration paths or in response to user prompts.

- It introduces a **hybrid object-centric scene structure**, rendering background regions as lightweight 2D surfaces while modeling foreground objects in full 3D to enrich user interaction. Accordingly, NeoWorld incorporates cutting-edge *differentiable rendering*, *representation learning*, and *image-to-3D reconstruction* techniques to create a unified world generation pipeline.

- Building on these features, NeoWorld enables new interactive capabilities not available in prior work, including **3D-consistent scene exploration** and **physics-based object manipulation**.

## 2 PRELIMINARIES

**Interactive world generation.** This task aims to construct a coherent sequence of spatially and semantically connected 3D scenes $\{\mathcal{E}_0, \mathcal{E}_1, \ldots\}$ starting from a single input image $I_0$, controlled by user-specified content prompts $P_i$ and camera trajectories $C_i$. This task involves two main stages that operate in an iterative *reconstruction-then-generation* manner:

- *Reconstruction*: At each time step $i$, a 3D scene representation $\mathcal{E}_i$ is generated from the current observation image $I_i$ using an *image-to-3D* module: $\mathcal{E}_i \sim \mathcal{M}_{3D}(I_i)$, where $\mathcal{M}_{3D}$ denotes a model that lifts 2D observations to explicit 3D scene representations.

- *Generation*: Based on the current scene representation $\mathcal{E}_i$, a user-defined camera movement $C_{i+1}$, and a text description $P_{i+1}$ of the new observation, the system synthesizes the next-view image: $I_{i+1} \sim \mathcal{G}(\mathcal{E}_i, C_{i+1}, P_{i+1})$, where $\mathcal{G}$ is an image synthesis model.

This iterative process allows the virtual world to progressively unfold as the user explores it, while maintaining spatial and temporal consistency.

**Existing methods and challenges.** Recent approaches such as WonderJourney (Yu et al., 2024) and WonderWorld (Yu et al., 2025) typically follow a two-step computation scheme for interactive world generation. First, user interactions or scripted camera paths determine the exploration trajectory. Then, generative inpainting models synthesize novel views conditioned on prior observations. The

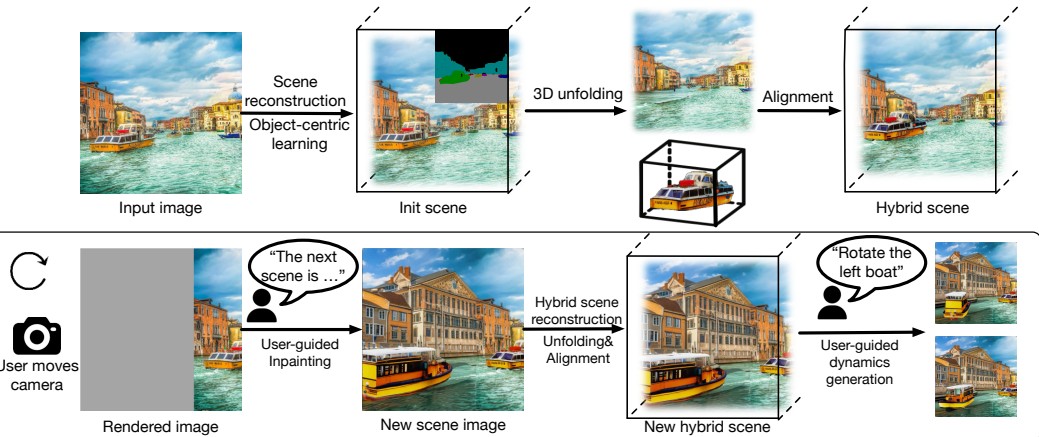

Figure 2: **The model architecture for 3D-consistent generation of physical worlds:** (i) an object-centric representation module, (ii) a progressive object-to-3D unfolding module, and (iii) a user interface that interprets natural language commands and drives simulation based on the 3D scene.

synthesized images are projected into 3D representations (*e.g.*, point clouds, meshes, or simplified 2.5D FLAGS (Yu et al., 2025)) and integrated into the existing environment, enabling the incremental construction of large-scale virtual worlds. However, these methods face several key limitations:

- *Limited interactions*: Existing methods primarily support visual navigation but lack support for physical interactions or dynamic animation. Without explicit object-centric modeling, fine-grained interaction with the generated world remains challenging.

- *Efficiency bottleneck in immersive 3D modeling*: Full-scene 3D generation is computationally expensive. While layered 2.5D representations (*e.g.*, FLAGS in WonderWorld (Yu et al., 2025)) offer higher efficiency, they inherently restrict the range of valid viewing angles. As a result, large viewpoint shifts often lead to geometric distortions or occlusion artifacts in the generated content.

## 3 METHOD

### 3.1 OVERVIEW

To tackle the aforementioned challenges, we propose NeoWorld, a unified framework that progressively constructs an open-ended interactive world from a single input image through an iterative *3D-unfolding-2D-generation* pipeline. Beyond visual navigation, NeoWorld focuses on object-centric world generation that is both efficient and immersive, and supports intuitive user–world interaction. An overview is shown in Fig. 2. Given a single input image, the scene is first reconstructed into object-centric Gaussian layers (2.5D) using panoptic segmentation. Key foreground objects are then reconstructed in full 3D, determined by predefined foreground categories and their distance to the camera. In this way, the scene is represented in a hybrid structure that combines object-centric 2.5D backgrounds with fully 3D foregrounds. This design offers two advantages: (i) balancing immersion and computational efficiency, and (ii) enabling object-level interaction with the generated world. As the user navigates or interacts with the scene, the system incrementally unfolds new regions of the world, guided by camera motion and user prompts. User commands—such as object manipulation or text-driven dynamics—are grounded in the generated entities; if the selected entity is in 2.5D, it will be reconstructed into 3D, thereby enabling interactive control and physically plausible animation. As an optional post-processing step for user–object interaction, we employ a video-to-video approach (Jiang et al., 2025) to further improve visual realism and motion smoothness.

As stated in Sec. 3.2–3.4, NeoWorld introduces three key innovations: (i) an object-centric neural scene representation, (ii) a progressive 2.5D-to-3D scene unfolding mechanism prioritized by object proximity or user prompts, and (iii) a user–scene interaction module that enables intuitive object-level manipulation and physics-based animation within the constructed world.

### 3.2 OBJECT-CENTRIC GAUSSIAN LAYERS

To enable object-aware 3D world construction from a single image, NeoWorld adopts an object-centric scene representation that combines layered Gaussian Spaltting (Yu et al., 2025) with compact instance-aware features. Refer to WonderWorld, we decompose the input image $\mathbf{I}_i$ into two depth

layers—foreground, background—using depth edges and object segmentations: $\mathbf{I}_i = \{\mathbf{I}_{\text{fg}}^i, \mathbf{I}_{\text{bg}}^i\}$. Each layer is represented as a set of 2D Gaussian primitives: $\mathcal{E}_i = \{\mathcal{E}_{\text{fg}}^i, \mathcal{E}_{\text{bg}}^i\}$. Each primitive can be regarded as a degenerate 3D Gaussian with a compressed depth scale ($\epsilon$), which preserves surface fidelity while maintaining efficient rendering. Unlike WonderWorld, we enrich each Gaussian with a learnable *object-centric attribute coefficient* $\gamma_n \in \mathbb{R}^C$, which encodes instance-level semantics in a low-dimensional embedding space (detailed in the next paragraph). This yields an object-centric scene layout. We initialize Gaussians using estimated depth and surface normals (Yu et al., 2025) (See Appendix E), and optimize their parameters with the photometric reconstruction loss between the rendered and input image $\mathbf{I}_i$. For scene extrapolation, we render novel views from the optimized Gaussian layers and apply an image inpainting model to complete missing regions. By repeating the cohesive loop of scene decomposition, optimizing object-centric Gaussian layers, novel-view rendering and inpainting, NeoWorld incrementally grows the world: $\{\mathcal{E}_0, \mathcal{E}_1, \ldots\}$. Next, we describe how the 2.5D Gaussian layers are bound with the object-centric attribute coefficients $\gamma_n$.

**Efficient object-centric attribute binding.** To derive $\gamma_n$ for each Gaussian primitive, we apply an off-the-shelf panoptic segmentation model (Jain et al., 2023) $g_{\text{seg}}$ independently to the foreground and background layers: $[\mathbf{M}_{\text{fg}}^i, \mathbf{S}_{\text{fg}}^i] = g_{\text{seg}}(\mathbf{I}_{\text{fg}}^i)$ and $[\mathbf{M}_{\text{bg}}^i, \mathbf{S}_{\text{bg}}^i] = g_{\text{seg}}(\mathbf{I}_{\text{bg}}^i)$, where $\mathbf{M}^i \in \mathbb{R}^{H \times W \times K}$ denotes an instance-level segmentation mask assigning each pixel to one of $K$ distinct objects, $K$ is an assumed maximum number of objects in the scene, and $\mathbf{S}^i \in \mathbb{R}^K$ provides the associated semantic categories, which are later used in object selections. A naive approach is to define $\gamma$ as a $K$-dimensional one-hot vector corresponding to object IDs, enabling segmentation masks to be rendered as: $\widehat{\mathbf{M}}(\mathbf{u}) = \sum_{n \in \mathcal{S}(\mathbf{u})} T_n(\mathbf{u}) \cdot \alpha_n \cdot \gamma_n$ with $T_n(\mathbf{u}) = \prod_{m \in \mathcal{S}(\mathbf{u}), o_m < o_n}(1 - \alpha_m)$ for pixel $\mathbf{u}$, where $\mathcal{S}(\mathbf{u})$ denotes Gaussians projected onto $\mathbf{u}$, sorted by depth, and $\alpha$ denotes opacity. The attributes $\gamma_n$ can then be optimized by a cross-entropy loss between $\widehat{\mathbf{M}}$ and the ground-truth segmentation $\mathbf{M}$. However, in the context of infinite world generation, the total number of objects $K$ can be extremely large. To address this, we introduce a compact codebook $\mathbf{F} \in \mathbb{R}^{K \times C}$ with $C \ll K$, which significantly reduces memory and computation cost: $\mathbf{F} = \{\mathbf{f}_1, \mathbf{f}_2, \ldots, \mathbf{f}_K\}, \mathbf{f}_k \in \mathbb{R}^C, \|\mathbf{f}_k\|_2 = 1$. Each embedding vector is uniformly sampled from the unit sphere in $C$-dimensional space, and their pairwise cosine similarities are constrained below a threshold $\delta$ to ensure robust instance discrimination. After initialization, the codebook is kept fixed and shared globally across all scenes. We render predicted embeddings $\gamma$ into segmentation space $\widehat{\mathbf{M}}$ and optimize them by minimizing the cosine distance to the codebook-augmented ground truth $\mathbf{M} \cdot \mathbf{F}$:

$$\mathcal{L}_{\text{cos}} = 1 - \frac{1}{|\Omega|} \sum_{\mathbf{u} \in \Omega} \frac{\widehat{\mathbf{M}}(\mathbf{u})^\top (\mathbf{M} \cdot \mathbf{F})(\mathbf{u})}{|\widehat{\mathbf{M}}(\mathbf{u})| \cdot |(\mathbf{M} \cdot \mathbf{F})(\mathbf{u})|}, \tag{1}$$

where $\Omega$ denotes the set of valid pixels. During initialization, Gaussian attributes are associated with codebook vectors according to 2D instance labels. At inference time, the instance label for a pixel $\mathbf{u}$ is predicted by selecting the nearest codebook vector: $y(\mathbf{u}) = \arg\max_{k \in 1, \ldots, K} \frac{\widehat{\mathbf{M}}(\mathbf{u})^\top \mathbf{f}_k}{|\widehat{\mathbf{M}}(\mathbf{u})| \cdot |\mathbf{f}_k|}$. This compact embedding strategy provides efficient and scalable feature encoding, making object-centric Gaussian representations feasible for infinite 3D world generation.

**Optimization.** The object-centric Gaissian layers are optimized by minimizing $\mathcal{L} = 0.8\mathcal{L}_1 + 0.2\mathcal{L}_{\text{D-SSIM}} + \mathcal{L}_{\text{cos}}$, where $\mathcal{L}_1$ and $\mathcal{L}_{\text{D-SSIM}}$ denote L1 and SSIM losses between the rendered and input image $\mathbf{I}_i$, and $\mathcal{L}_{\text{cos}}$ measures the cosine distance between $\gamma$ and $\boldsymbol{f}$. To further promote spatial smoothness of object-centric representations, we periodically replace each $\gamma$ with the mean value of its $k$-nearest neighbors during training (KNN smoothing). This strategy effectively suppresses floaters (*i.e.*, outlier Gaussians) and enhances overall geometric consistency across the scene.

**Cross-scene alignment.** A key challenge is ensuring that object-centric Gaussian layers maintain instance-level continuity across different viewpoints. To address this, we establish correspondences between the newly obtained panoptic masks and the previously predicted instance labels. Given a panoptic segmentation mask $\mathbf{M}^i$ at the current viewpoint $\boldsymbol{C}_i$ and the predicted instance label map $y_{i-1}$ rendered from the prior scene representation, we perform correspondence matching within the overlapping regions. Specifically, each current panoptic instance $k$ is re-assigned to the predicted label $y_{i-1}$ if their overlapping area exceeds a predefined threshold $\theta$. This matching procedure enables consistent label propagation across views, ensuring that the object-centric attributes $\gamma$ attached to each Gaussian remain coherent as the scene evolves. Therefore, NeoWorld constructs a continuous object-centric representation for incrementally expanding environments.

## 3.3 Progressive 2.5D-to-3D Unfolding

Although object-centric Gaussian layers are efficient, they are not well-suited for interactions such as object manipulation and animation. Meanwhile, 2.5D layers often introduce noticeable artifacts under extreme viewpoint changes. Therefore, it is essential to reconstruct interaction-relevant objects with full 3D geometry. In particular, since foreground objects are the most likely to involve interactions, we prioritize those belonging to predefined foreground categories and located closest to the current viewpoint, selecting the top $N$ objects by proximity. In such cases—or when explicitly specified by user prompts—we invoke an image-to-3D module (Amodal3R (Wu et al., 2025) in practice) for object completion and alignment (Sec. 3.3).

**3D object alignment.** Reconstructed 3D objects are often misaligned in position, rotation, or scale relative to the existing Gaussian layers $\mathcal{E}_i$ and the object's original placement. To seamlessly integrate them into the scene, we perform alignment by optimizing uniform scale $S \in \mathbb{R}^+$, rotation $\mathbf{R} \in \mathbb{R}^{3\times3}$, and translation $\mathbf{T} \in \mathbb{R}^3$. This procedure consists of two stages. (1) **Coarse alignment.** Prior work typically searches over a discrete set of yaw, pitch, and roll angles and selects the best hypothesis via a perceptual metric (e.g., DINOv2) (Hu et al., 2025). This approach is computationally expensive due to the large candidate set and repeated perceptual evaluations. Instead, we leverage the priors of an image-to-3D reconstruction model and fine-tune it to jointly diffuse object geometry and pose. Concretely, we fine-tune the *Sparse Structure Transformer* of the Amodal3R, and augment the DiT input with an additional pose token $\mathcal{E}(p)$, where $p \in \mathbb{R}^6$ is a 6D rotation parameterization. During training, the ground-truth pose $p^\star$ is perturbed along a flow-matching path $p_t$ and fed to the DiT, which predicts velocity fields for both geometry and pose under a flow-matching objective. At inference, we sample $p_T \sim \mathcal{N}(0, I_6)$ and integrate the reverse flow to obtain $p_0$. The 6D rotation is mapped to $\mathrm{SO}(3)$ via Gram–Schmidt. Scale $S$ is initialized by matching the longest edge of the reconstructed bounding box to the target, and translation $\mathbf{T}$ aligns centers. Since our method adds only one token, pose estimation incurs negligible overhead compared to the base image-to-3D pipeline. (2) **Fine alignment.** We further refine translation, scale, and rotation by minimizing a differentiable rendering objective on the original scene. Specifically, we employ a depth loss and a silhouette Dice loss between renderings of the reconstructed object and the ground-truth target, ensuring precise alignment and seamless integration.

**Fallback for unreliable 3D reconstruction.** Although recent advances in image-to-3D reconstruction (Wu et al., 2025; Xiang et al., 2024; Yushi et al., 2025) have demonstrated strong performance, errors may still arise, particularly when object segmentation is inaccurate under occlusion. To enhance the robustness of NeoWorld, we introduce a fallback strategy: after unfolding and aligning the object to the input image, we evaluate reconstruction fidelity by computing the cosine similarity between DINOv2 features of the re-rendered object and its corresponding masked region in the input. If the similarity score falls below a threshold $\tau$, the object is reverted to a 2.5D representation, as low similarity typically reflects segmentation errors or degraded 3D reconstruction under severe occlusion. Additional ablation details are provided in Appendix B.

## 3.4 Intuitive User-World Interaction

Recall that the generated world is object-centric, consisting of 3D foreground objects and object-centric Gaussian layers. We further enable user prompts to manipulate or animate arbitrary objects within the world. To achieve this, we employ a Large Language Model ($g_{\text{LLM}}$, Gemini-2.5pro (Comanici et al., 2025)) to interpret user intent. The input to $g_{\text{LLM}}$ is decomposed into three components: the instruction $\mathcal{J}$ (defining scene interaction rules), the user prompt $\mathcal{U}$ (specifying the desired manipulation), and $\mathcal{O}$ (describing all scene objects by their spatial centers, scales, and categories). Given these inputs, $g_{\text{LLM}}$ predicts the target object index $\mathcal{I}$ and the corresponding manipulation attributes $\mathcal{A}$: $[\mathcal{I}, \mathcal{A}] = g_{\text{LLM}}(\mathcal{J}, \mathcal{O}, \mathcal{U})$. Examples and further implementation details are provided in Appendix E. The attributes $\mathcal{A}$ are task-dependent and may include translations and rotations for basic manipulations, transformation sequences for animations (*e.g.*, lists of translations and rotations), or physical parameters (*e.g.*, material properties for MPM-based dynamic simulation). To support more complex interactions, we further allow objects to be converted into meshes or substituted with high-fidelity 3D assets. These assets can then be animated using keyframe techniques, thereby enhancing both realism and immersion in interactive world generation.

**Video-to-Video enhancement.** While MPM-based simulation and animations can produce physically plausible and 3D-consistent dynamics, they still have important limitations: in particular, they cannot adequately handle appearance changes induced by object–environment interactions, such as

moving shadows or water flowing and splashing as a boat moves. To further enhance the realism of the scene, we leverage a state-of-the-art video-to-video (V2V) generation model (Jiang et al., 2025) to refine the simulated dynamics, yielding visually higher-quality and more coherent dynamic videos. To enable a fair and transparent comparison with the baselines, all reported results are obtained **without** applying the visual enhancement module, unless otherwise specified.

## 4 EXPERIMENTS

### 4.1 EXPERIMENTAL SETUP

**Implementation details.** Following WonderWorld, we use StableDiffusion-v2.0-Inpainting (Rombach et al., 2022) as the backbone for inpainting and distilled StableDiffusion-XL for object removal. For panoptic segmentation, we adopt OneFormer (Jain et al., 2023). Normal and depth estimation are performed with Marigold Normal and Marigold Depth (Ke et al., 2024) to ensure high-quality geometric information. For scene alignment, we fine-tune Amodal3R for 20 epochs on a mixture of 3D synthetic datasets: 3D-FUTURE (Fu et al., 2021), ABO (Collins et al., 2022), and HSSD (Khanna et al., 2024). Hyperparameters are set as follows: codebook dimension $C = 16$, cosine similarity threshold $\delta = 0.5$, and fallback score threshold $\tau = 0.4$. We sample 3 viewpoints along the fixed panoramic path and 15 additional viewpoints at $30°$ intervals on the orbiting path. All images are rendered at $512 \times 512$ resolution with evenly spaced viewpoints.

**Baselines.** Since no prior work supports interactive 3D object-centric world generation, we perform best-effort comparisons with three groups of baselines, each targeting a different aspect of NeoWorld.

- *Unbounded world generation*: We compare with recent 3D world generation methods (Wonder-Journey (Yu et al., 2024), WonderWorld (Yu et al., 2025)), video diffusion models (CogVideoX-I2V-5B (Hong et al., 2023), Wan2.1-I2V-14B (Wan et al., 2025)), and Matrix-Game2 (He et al., 2025), an interactive 2D world generation baseline.
- *Object-centric accuracy*: We evaluate against 3D object-centric learning methods, Gaussian-Grouping (Ye et al., 2024) and OmniSeg3DGS (Ying et al., 2024). GaussianGrouping distills 3D segmentations from 2D masks (SAM (Kirillov et al., 2023), DEVA (Cheng et al., 2023)), while OmniSeg3DGS learns 3D feature fields from SAM masks via contrastive learning (Li et al., 2020).
- *Interactive manipulation*: As ground-truth 3D dynamics are unavailable, we compare with strong video models (Kling 1.6 (Kuaishou, 2025), CogVideo-I2V, Wan2.1-I2V) and PhysGen3D (Chen et al., 2025), which targets physics-plausible world dynamics.

**Benchmarks.** We construct our evaluation benchmark following three prior works: WonderWorld, WorldScore (Duan et al., 2025), and WonderJourney. To ensure consistency, we exclude wide-angle landscape photos with vast scenery or ambiguous composition, resulting in a curated set of 28 images covering 7 distinct styles and occlusion conditions. Following the automatic evaluation protocol of WonderWorld, we procedurally generate 4 3D environments per image, yielding 112 diverse scenes spanning both photorealistic and artistic styles. Scene descriptions are produced using ChatGPT (Achiam et al., 2023), and the camera trajectory is fixed to a panoramic path (see WonderWorld for procedural generation details). For novel-view evaluation, we additionally adopt an orbiting trajectory with azimuth sweeping from $0°$ to $90°$, inspired by WorldScore.

**Metrics.** Following prior work (Yu et al., 2025; Duan et al., 2025), we evaluate **static world** generation and novel-view exploration using the following metrics:

1. *CIQA+* (Wang et al., 2023), *Q-Align* (Wu et al., 2024a), and *sFID* (Nash et al., 2021) to assess perceptual and semantic image quality compared with real data;
2. *3D Consistency* and *Scene Quality* measured by human users for scene realism and overall video quality along generation and exploration trajectories;
3. *ImageCLIP* for text-scene alignment and *CLIP Score* for long-term consistency between the input image and novel views;
4. *IoU* for segmentation accuracy against ground-truth masks;
5. Additionally, we report three *VBench* (Zhang et al., 2024a) metrics for video quality evaluation, including motion smoothness, subject consistency, and background consistency.

For **dynamic world** generation, such as multi-object scenarios with spatially grounded prompts (*e.g.,* "the chair on the left"), which require precise object identification and animation, we evaluate two metrics: (1) *Prompt Alignment*, a human study measuring text–video alignment, and (2) *VideoCLIP Similarity*, an automated score computed with VideoCLIP-XL (Wang et al., 2024).

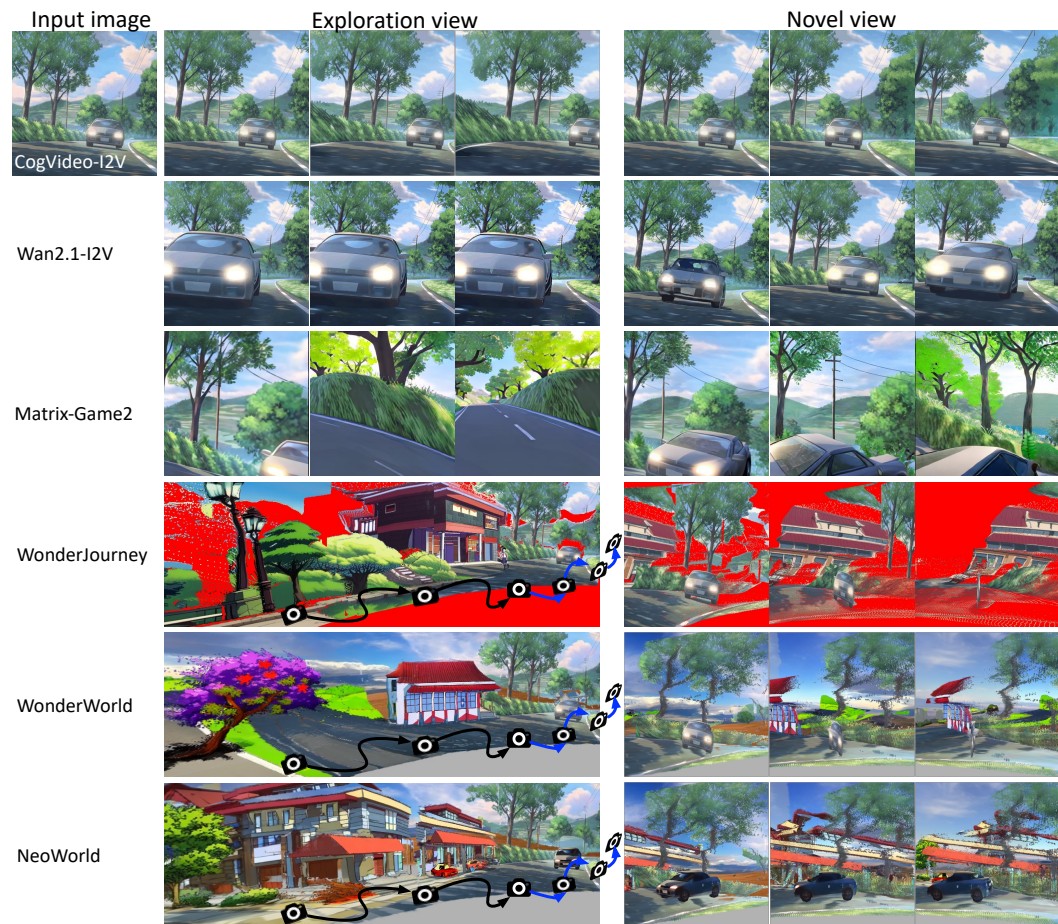

Figure 3: **Qualitative comparison of exploration view and novel view rendering**. Camera viewpoints follow the illustrated trajectory, with the novel view path shown in blue.

Table 1: **Interactive world generation performance.** Human evaluation results are indicated with †. The time required to generate each novel view is measured on an NVIDIA H20 GPU. For all metrics except time cost, higher values indicate better performance.

| Method | CIQA+ | Q-Align | 3D-Const† | SceneQuality† | ImageCLIP | CLIP-Score | Time/view (s) |
|---|---|---|---|---|---|---|---|
| CogVideo-I2V | 0.65 | 4.09 | N/A | N/A | **76.23** | 92.47 | 242.53 |
| Wan2.1-I2V | **0.67** | **4.28** | N/A | N/A | 74.54 | **95.43** | 721.20 |
| Matrix-Game2 | 0.58 | 3.76 | N/A | N/A | N/A | 70.36 | **8.57** |
| WonderJourney | 0.49 | 1.73 | 20.33 | 20.51 | **78.91** | 66.00 | 179.11 |
| WonderWorld | 0.55 | 2.34 | 32.42 | 32.26 | 78.35 | 69.20 | **10.71** |
| **NeoWorld** | **0.59** | **2.66** | **47.25** | **47.23** | 78.63 | **72.46** | 18.14 |

## 4.2 EVALUATION ON UNBOUNDED WORLD GENERATION

In Fig. 3, we present a qualitative comparison of exploration-view and novel-view renderings across NeoWorld, CogVideo-I2V, Wan2.1-I2V, Matrix-Game2, WonderWorld, and WonderJourney. We can see that only NeoWorld can keep 3D view realism without explicit holes, benefiting from its hybrid scene representation. More showcases are included in the Appendix D. Table 1 reports results of NeoWorld against two 3D world generation methods (WonderJourney, WonderWorld) and three video diffusion models (CogVideo-I2V, Wan2.1-I2V, Matrix-Game2). Additionally, Table 2 presents quantitative results on supplementary metrics. NeoWorld achieves the lowest sFID and the highest VBench scores among all methods, demonstrating superior visual fidelity, smoother motion, and more consistent temporal behavior overall.

Table 2: Quantitative comparison of interactive world generation in **sFID** and three **VBench** (Zhang et al., 2024a) metrics, including motion smoothness, subject consistency, and background consistency.

| Method | sFID↓ | Motion Smoothness↑ | Subject Consistency↑ | Background Consistency↑ |
|---|---|---|---|---|
| WonderJourney | 114.17 | 0.9810 | 0.7741 | 0.8806 |
| WonderWorld | 111.07 | 0.9886 | 0.7895 | 0.8819 |
| NeoWorld | **109.54** | **0.9897** | **0.7917** | **0.8840** |

Table 3: **Quantitative analysis of the proposed object-centric representation (Metric: IoU).**

| OmniSeg3DGS | GaussianGrouping | NeoWorld | w/o Joint Optim. | w/o KNN Smooth |
|---|---|---|---|---|
| 33.24 | 36.70 | **70.53** | 64.26 | 68.59 |

**3D scene realism.** We evaluate 3D consistency (3D-Const) and overall scene quality (SceneQuality) through a human study comparing WonderJourney, WonderWorld, and NeoWorld. Over $45\%$ of participants preferred NeoWorld. Video diffusion models are excluded as they do not support 3D world generation or accurate viewpoint control. On CIQA+ and Q-Align, Wan2.1-I2V and CogVideo-I2V achieve higher scores due to minimal camera motion and limited viewpoint changes, producing frames that closely match the input images. Nevertheless, NeoWorld surpasses WonderJourney and WonderWorld on both metrics, demonstrating stronger visual realism in interactive 3D generation.

**Text-to-scene alignment and long-term consistency.** NeoWorld achieves a comparable Image-CLIP score to WonderJourney and WonderWorld, while diffusion-based methods show markedly lower text-to-scene similarity, reflecting weaker geometric grounding. Matrix-Game2 is excluded from ImageCLIP as it lacks text input. For temporal coherence, NeoWorld attains the highest CLIP score among Matrix-Game2, WonderJourney, and WonderWorld; Wan2.1-I2V and CogVideo-I2V score higher because near-static cameras inflate frame-level similarity without true 3D consistency.

**Efficiency.** NeoWorld attains the second-best rendering speed among 3D unbounded world generation methods. Its efficiency mainly stems from the progressive 3D unfolding procedure, despite incorporating object-centric learning and object-to-3D generation. Overall, NeoWorld offers the best balance of realism, exploration, and efficiency.

### 4.3 EVALUATION ON OBJECT-CENTRIC REPRESENTATIONS

We manually annotated instance-level masks as ground truth and computed the IoU against the rendered masks. Quantitative results are reported in Table 3. Even without joint optimization or KNN smoothing (see Sec. 3.2), NeoWorld significantly outperforms OmniSeg3DGS and GaussianGrouping. When jointly optimized with image reconstruction loss ($\mathcal{L}_1$ and $\mathcal{L}_{\text{D-SSIM}}$) and object-centric loss $\mathcal{L}_{\cos}$, the IoU improves from $64.26$ to $70.53$, demonstrating the benefit of leveraging implicit correlations between appearance and instance semantics. Applying KNN smoothing further suppresses Gaussian floaters, increasing IoU from $68.59$ to $70.53$. Qualitative comparisons in Fig. 4 show that the instance masks generated by NeoWorld align more accurately and smoothly with the RGB images than those of OmniSeg3DGS, further validating the effectiveness of our object-centric representation.

### 4.4 EVALUATION ON USER INTERACTIONS

By leveraging the parsing capabilities of LLMs, NeoWorld enables user-prompt–controlled object manipulation and animation. As shown in Fig. 5, given prompts such as "rightmost boat" or "right chair," the manipulation targets are correctly located and animated. Compared with strong video diffusion models, including CogVideo-I2V (Hong et al., 2023), Wan2.1-I2V (Wan et al., 2025), and the commercial Kling1.6 (Kuaishou, 2025), NeoWorld achieves superior text–motion

Table 4: **Interactive dynamic world animation performance.** Higher values indicate better performance. Similarly, human evaluation results are indicated with †.

| Method | PromptAlign[†] | VideoCLIP | Method | PromptAlign[†] | VideoCLIP |
|---|---|---|---|---|---|
| CogVideo-I2V | 8.63 | 16.34 | WonderJourney | N/A | N/A |
| Wan2.1-I2V | 8.52 | 16.26 | WonderWorld | N/A | N/A |
| Kling 1.6 | 20.90 | 16.19 | **NeoWorld** | **61.95** | **17.05** |

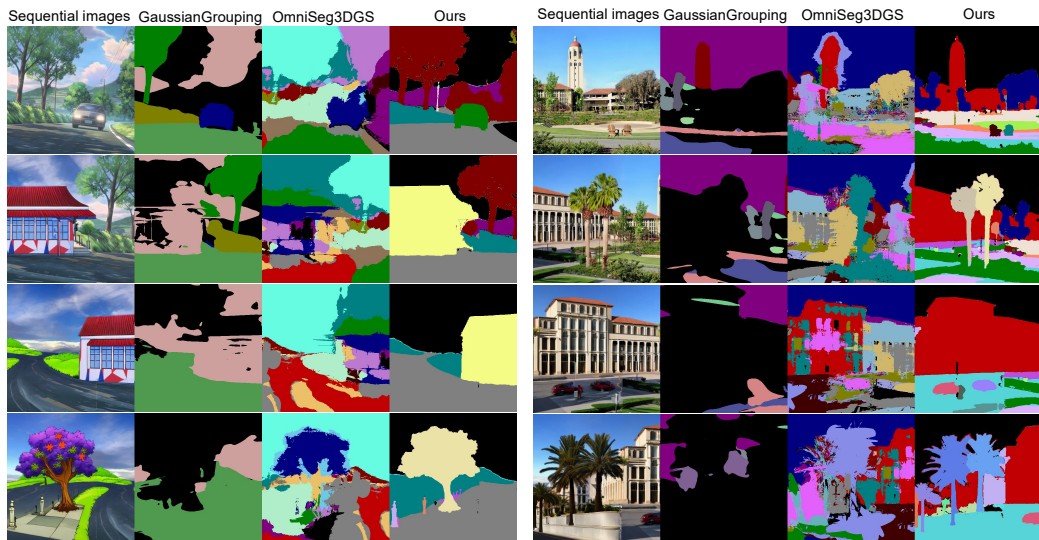

Figure 4: **Qualitative comparison of object-centric representation**.

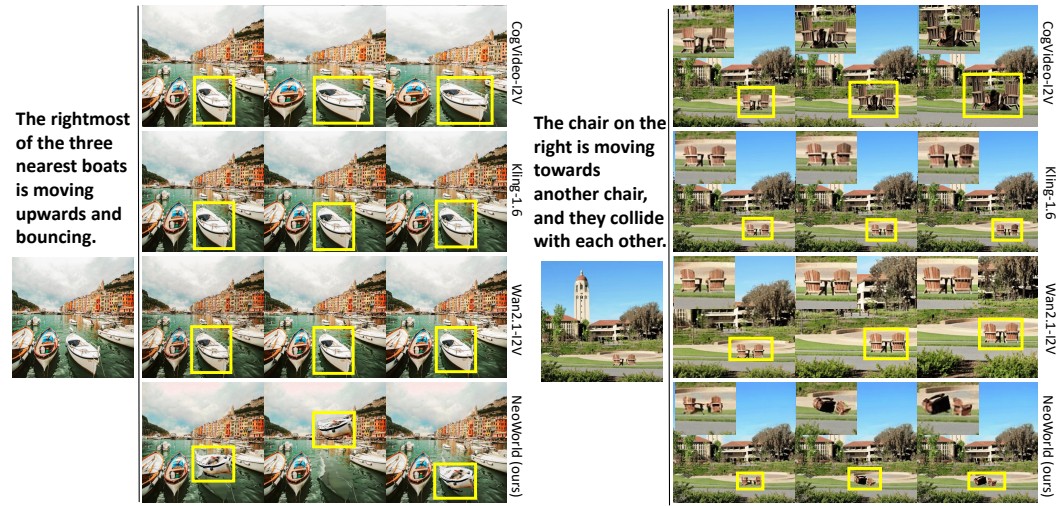

Figure 5: **Qualitative results of dynamic simulation**.

alignment. Quantitative results in Table 4 confirm this: both human study results (PromptAlign) and VideoCLIP scores demonstrate the effectiveness of NeoWorld in aligning generated dynamics with user instructions. In contrast, previous interactive 3D world generation models (WonderJourney and WonderWorld) are not object-centric; they support only visual navigation and cannot enable text-guided object control. Due to space limitations, we refer readers to Appendix B, D for additional examples of object manipulation and further analysis of LLM design and behavior.

### 4.5 ABLATION STUDIES

**Post-simulation visual enhancement.** In Fig. 6, we further present dynamic simulation and animation results *with* and *without* the visual enhancement module. The results indicate that the post-V2V module significantly improves the overall image quality and scene coherence, producing natural appearance changes caused by interactions between objects and the environment, including evolving lighting and shadows, as well as water flickering and rippling.

**Alternative module designs.** We conduct ablation studies on key backbone choices in NeoWorld, including depth/normal estimators (Bhat et al., 2023; Xu et al., 2025; Bae & Davison, 2024), in-painting models (Suvorov et al., 2021; Labs, 2024), and image-to-3D models (Szymanowicz et al., 2024). As shown in Table 5, lighter models already yield reasonable performance, while stronger ones (*e.g.*, PPD for depth and Marigold for normals)consistently provide improvements. LaMa yields

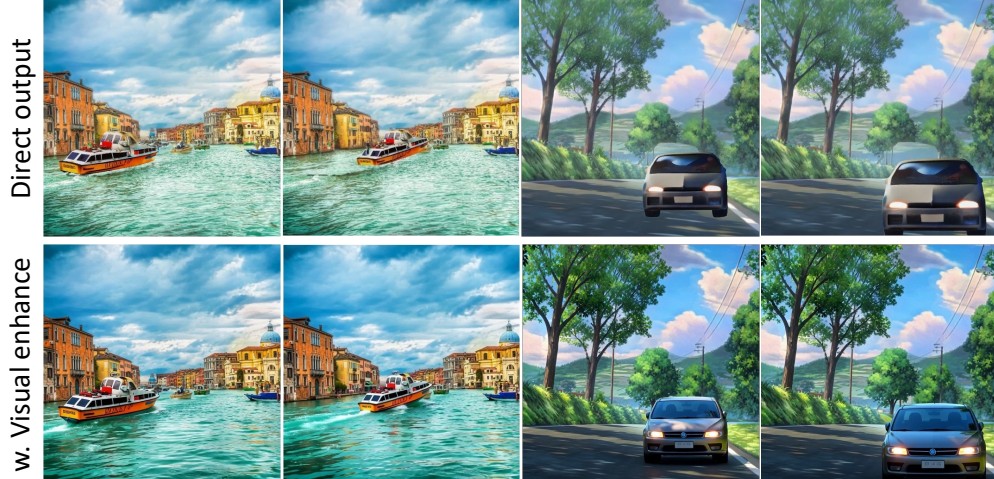

Figure 6: **Showcases of dynamic simulation *without* and *with* the post visual enhancement module.** The final version exhibits more natural visual effects, such as water ripples following the boat, realistic shadows of the car cast on the ground, and a reduced floating appearance of the car.

Table 5: **Ablation study of module designs in NeoWorld.** Compared with the modules used in NeoWorld, we denote lighter alternatives with [†] and stronger alternatives with [‡].

| Task | Model | CIQA+ | Q-Align | ImageCLIP | CLIP-Score |
|---|---|---|---|---|---|
| Depth | replace w. ZoeDepth[†] | 0.58 | 2.64 | 77.57 | 73.96 |
| | replace w. PPD[‡] | 0.58 | 2.78 | 78.18 | 74.37 |
| Normal | replace w. DSINE[†] | 0.48 | 2.56 | 77.51 | 70.33 |
| Inpainting | replace w. LaMa[†] | 0.52 | 2.53 | 54.95 | 66.80 |
| | replace w. SD1.5[†] | 0.57 | 2.63 | 76.67 | 70.30 |
| | replace w. Flux-Fill[‡] | 0.56 | 2.65 | 76.11 | 69.10 |
| Image-to-3D | replace w. SplatterImage[†] | 0.54 | 2.55 | 78.44 | 68.43 |
| Final | NeoWorld | 0.59 | 2.66 | 78.63 | 72.46 |

lower ImageCLIP scores because it is text-agnostic, and Flux-Fill does not produce further gains since it is designed for local object replacement rather than the large-scale completion required in NeoWorld. Additional ablations, including LLM choice, object removal, codebook design, alignment, and hyperparameters, are provided in Appendix B.

## 5 CONCLUSIONS AND LIMITATIONS

In this work, we introduced NeoWorld, a novel deep learning framework for interactive world generation with object-level semantics and 3D physical consistency. In contrast to existing approaches that are constrained to static world generation and limited to visual navigation, NeoWorld enables user-driven object manipulation and physics-based dynamic simulation within a continuously expanding 3D environment. To achieve this, we designed a cascaded architecture that starts with lightweight 2D object-centric representations and progressively unfolds full 3D geometry based on user interactions, effectively balancing computational efficiency with immersive visual and physical realism.

Rather than a single unified model, NeoWorld is a cascade of external, pre-trained modules. Consequently, end-to-end robustness is constrained by the weakest link, and upstream errors can propagate to the final world simulation. Typical failures include: (i) alignment failures; (ii) ambiguous or overly complex prompts that lead to LLM misinterpretation; (iii) image-to-3D reconstruction errors under heavy occlusion or highly complex/reflective textures; and (iv) under- or over-segmentation results, which corrupt object masks and the following reconstruction. **Please refer to the Appendix F for detailed analyses and visualizations.**

## REPRODUCIBILITY STATEMENT

We include anonymized code in the supplementary material to facilitate the reproduction of all experiments, figures, and tables. The Implementation Details section in the appendix specifies all hyperparameter settings. We will release a de-anonymized repository upon acceptance.

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

## APPENDIX

This supplementary material includes the following:

- *Related work*: Introduction of related direction, including infinite world generation and object-level 3D scene decomposition.

- *Ablation study*: Ablations of key components, including codebook design, object removal, LLM choice, alignment, multi-object manipulation, scalability analysis, and other hyperparameters (Sec. B).

- *Quantitive results*: Detailed benchmark description and quantitive results (Sec. C).

- *Qualitative results*: Additional visualizations of generated scenes and simulations, including interactive world generation results using exploration views only and using both exploration and novel views, simulations with multiple objects, and multi-view renderings (Sec. D).

- *Further Implementation details*: Additional information on Gaussian layer initialization, human study setup, per-module time breakdown, and prompt design for LLMs (Sec. E).

- *Failure case analysis*: Visualizations and analysis of typical failure cases (Sec. F).

## A  RELATED WORK

### A.1  INFINITE WORLD GENERATION

Infinite world generation aims to construct an unbounded world from a single image, enabling real-time control via camera motion and content prompts. Early research focused on perpetual video generation along a given camera trajectory. The seminal work InfiniteImages (KANEVA et al., 2010) introduced a non-parametric method for infinite 2D extrapolation through classical 2D image retrieval, stitching, and blending. Subsequent learning-based methods (Liu et al., 2021; Lin et al., 2022; Li et al., 2022b; Cai et al., 2023; Chai et al., 2023; Raistrick et al., 2023; Bruce et al., 2024; Yang et al., 2024b; Feng et al., 2024; Raistrick et al., 2024; Zhou et al., 2025; Ni et al., 2025) auto-regressively synthesized new scenes with generative models (Zhuang et al., 2024; Karras et al., 2019; Rombach et al., 2022; Song & Ermon, 2019; Podell et al., 2024; Ke et al., 2024). Recent advances have extended from 2D to 3D scene exploration (Hu et al., 2021; Yu et al., 2024; Fridman et al., 2023; Yu et al., 2025; Höllein et al., 2023; Lu et al., 2024; Zhang et al., 2024b; Lin et al., 2023) by integrating image-to-3D generation (Xiang et al., 2024; Wu et al., 2024b; Hong et al., 2024; Yushi et al., 2025; Wu et al., 2025) after the image extrapolation step. Wonderworld (Yu et al., 2025) even realized real-time performance through the proposed efficient 2.5D layered scene representation. However, existing methods remain limited to view-controlled navigation, lacking support for fine-grained user-world interactions like physical manipulation or dynamic animation.

### A.2  OBJECT-LEVEL 3D SCENE DECOMPOSITION

2D scene decomposition (Greff et al., 2016; 2019; Burgess et al., 2019; Engelcke et al., 2020; Elsayed et al., 2022; Kipf et al., 2022; Singh et al., 2022; Xie et al., 2022) typically uses open-vocabulary segmentation (Zhang et al., 2023; Qin et al., 2023; Zhu & Chen, 2024; Liu et al., 2024b) or unsupervised methods like slot attention (Locatello et al., 2020). For 3D, recent works (Qiu et al., 2024; Zhao et al., 2025; 2024; Kabra et al., 2021; Sajjadi et al., 2022; Chen et al., 2021; Driess et al., 2023; Yang et al., 2024a; Luo et al., 2024; Qin et al., 2024; Kobayashi et al., 2022; Tschernezki et al., 2022; Siddiqui et al., 2023; Kerr et al., 2023) attach semantics into neural fields (Mildenhall et al., 2020; Kerbl et al., 2023) by distilling features from models (*e.g.*, CLIP (Radford et al., 2021), DINO (Caron et al., 2021; Oquab et al., 2024), LSeg (Li et al., 2022a), or SAM (Kirillov et al., 2023; Ravi et al., 2025)), across multiple viewpoints. There are also some efforts (Kohli et al., 2020; Stelzner et al., 2021; Zhi et al., 2021; Liu et al., 2023) that leverage direct supervision (*e.g.*, depth or instance maps). However, current approaches require dense views and suffer from high training or optimization costs. The key challenge remains: online semantic reconstruction from sparse (even monocular) input.

Table 6: **Comparison of alternative designs for object-centric representations.** These results are achieved on 9 scenes using 3 different seeds. Our codebook design yields a great balance between the object-centric scene decomposition quality and rendering efficiency. *Time* denotes the average training time for a single scene layer, and *Storage* denotes the storage required for a world consisting of 9 scenes.

| Method | IoU | Time (s/scene) | Storage |
|---|---|---|---|
| One-hot Encoding | **92.16 ± 1.92** | 52.90 | 2726M |
| AutoEncoder | 24.42 ± 2.40 | 2.59 | 334M |
| Linear Mapping | 45.54 ± 4.60 | 3.95 | 333M |
| **Codebook (Final model)** | 86.27 ± 1.23 | **2.54** | 333M |

Table 7: **Comparison of difference alignment strategies**. Plausibility and coherence are evaluated through a human-in-the-loop study. Our approach achieves the best overall alignment performance while maintaining reasonable efficiency.

| Method | Plausibility | Coherence | Time(s) |
|---|---|---|---|
| w/o Coarse | 10.71 | 11.20 | 1.86 |
| w/o Fine | 25.67 | 26.17 | **0.06** |
| Flash Sculptor (Hu et al., 2025) | 29.75 | 30.18 | 105.06 |
| Full model | **33.87** | **32.45** | 1.92 |

## B ABLATION STUDY

**Alternative designs for object-centric representations.** As discussed in Sec. 3.2, a straightforward approach for object-centric learning is to define $\gamma$ as a $K$-dimensional one-hot vector, which directly corresponds to object IDs. Additionally, prior work has proposed alternative designs, such as employing an autoencoder to first compress feature vectors into a lower-dimensional space (Qin et al., 2024), or utilizing a single linear layer to map the rendered feature map from a lower-dimensional space back to its original high-dimensional representation (Ye et al., 2024).

We report the IoU, the average training time for a single scene layer (*e.g.*, $\mathcal{L}_{fg}$), and the storage for a world consisting of 9 scenes in Table. 6. From the results, it can be observed that one-hot encoding achieves the highest IoU, but at the cost of significantly higher training time and memory consumption. This makes it impractical for interactive infinity world generation, where computational efficiency is essential. In contrast, both the autoencoder and linear mapping achieve suboptimal results for different reasons.

The autoencoder suffers from the lack of explicit constraints on the distances of the compressed representations, leading to reduced robustness. On the other hand, linear mapping approaches are usually applied in offline settings, where the entire set of scenes is pre-defined and known beforehand. In our online scenario, where scenes are generated incrementally, linear mapping faces catastrophic forgetting issues. Furthermore, linear mapping requires projecting low-dimensional features into high-dimensional space for loss computation, which is notably slower compared to our approach, where cosine similarity is directly applied in the low-dimensional codebook.

Notably, different from Sec. 4.3, here we evaluate the performance using the IoU between the predicted labels and the panoptic mask generated by OneFormer (Jain et al., 2023). This metric provides a clearer and more intuitive way to reflect distillation errors. Overall, our method strikes a good balance between performance and efficiency, making it a suitable choice for infinite world generation under interactive scenarios.

**Ablation study of object alignment.** In Table 7 and Fig. 7, we ablate our alignment pipeline by: (i) removing coarse alignment, (ii) removing fine alignment, and (iii) replacing coarse alignment with Flash Sculptor (Hu et al., 2025), which performs a discrete search over predefined angles using DINOv2 similarity. We evaluate physical plausibility, visual coherence (via a human-in-the-loop study), and efficiency, where the reported time for coarse alignment is measured as the overhead relative to the original image-to-3D pipeline. The results show that our coarse alignment achieves strong alignment results with almost no additional time cost, and is critical for producing plausible

Table 8: **Sensitivity analyses**. We evaluate the impact of varying the cosine similarity threshold $\delta$ and the codebook dimension $C$ on the performance of object-centric representation learning. The results are derived from 9 scenes using 3 different seeds. *Time* denotes the average training time for a single scene layer, and *Storage* denotes the storage required for a world consisting of 9 scenes.

| Hyperparameters | IoU | Time(s/scene) | Storage |
|---|---|---|---|
| $\delta = 0.9$, $C = 8$ | $83.09 \pm 2.80$ | 2.28 | 257M |
| $\delta = 0.7$, $C = 11$ | $84.34 \pm 1.90$ | 2.40 | 287M |
| $\delta = 0.5$, $C = 16$ (Final model) | $86.27 \pm 1.23$ | 2.54 | 333M |
| $\delta = 0.3$, $C = 90$ | $87.24 \pm 1.50$ | 7.94 | 564M |

Table 9: **The impact of codebook size on object-centric representation learning.** The results are derived from 9 scenes. *Storage* denotes the storage required for a world consisting of 9 scenes.

| Codebook size | IoU | Storage |
|---|---|---|
| 16 | 18.71 | 409M |
| 128 | 84.08 | 409M |
| 384 | 85.88 | 409M |
| 16 / scene | 79.81 | 409M |
| 256 (Ours) | **87.03** | 409M |

and coherent outputs, while fine alignment further refines the results. Overall, our method delivers the highest alignment quality with substantially lower runtime than Flash Sculptor.

**Hyperparameter analyses.** In Table 8, we analyze the impact of two key hyperparameters: the codebook dimension $C$ and the cosine similarity threshold $\delta$. A higher threshold $\delta$ enables the use of a smaller codebook dimension $C$, improving computational efficiency. However, this comes at the expense of reduced robustness, as higher similarity thresholds may result in less distinct object representations. In this experiment, we tuned $\delta$ and adjusted $C$ to the minimum value that satisfies the threshold. In our final model, we set $\delta = 0.5$ and $C = 16$, achieving a favorable balance between efficiency and robustness. In Table 9, we evaluate the impact of codebook size on the performance of object-centric representation learning, and additionally compare a per-scene codebook variant. The results show that as long as the global codebook size is larger than the typical number of objects, the overall performance is very similar, and the codebook size is essentially irrelevant to the total world storage. In contrast, using a per-scene codebook leads to degraded performance: newly added scenes may introduce codebook entries that are similar to those of existing scenes, which increases feature ambiguity and results in noisy or incorrect segmentations.

**Ablation study of LLMs.** To constrain LLM outputs to be physically plausible and within a reasonable operating range, we augment the instruction prompt $\mathcal{J}$ with targeted selection guidance. As an alternative, we supply few-shot exemplars during prompting to encourage the LLM to produce more accurate, context-aware manipulation attributes. To quantify the effect of in-context learning on overall system performance, we conduct the following study. Specifically, we inject 4 exemplars into the prompt, each comprising a user instruction, relevant object metadata, and the expected outputs. The model is evaluated on 8 diverse scenes spanning a broad stylistic spectrum and both simple and complex cases. For comparison, we also evaluate a no-guidance baseline in which all attribute cues are removed from the prompt. We report quantitative results on three metrics:

- **Object selection accuracy:** We manually annotated the dataset comprising prompts and their corresponding target objects to evaluate whether the model accurately selects the intended object.

- **Motion alignment:** We conducted a human-in-the-loop study to assess whether the simulated or animated movements reflect the user's intent.

- **Penetration rate (for animation):** Similar to motion alignment, we employed a human-in-the-loop study to evaluate whether objects exhibit unnatural interpenetration.

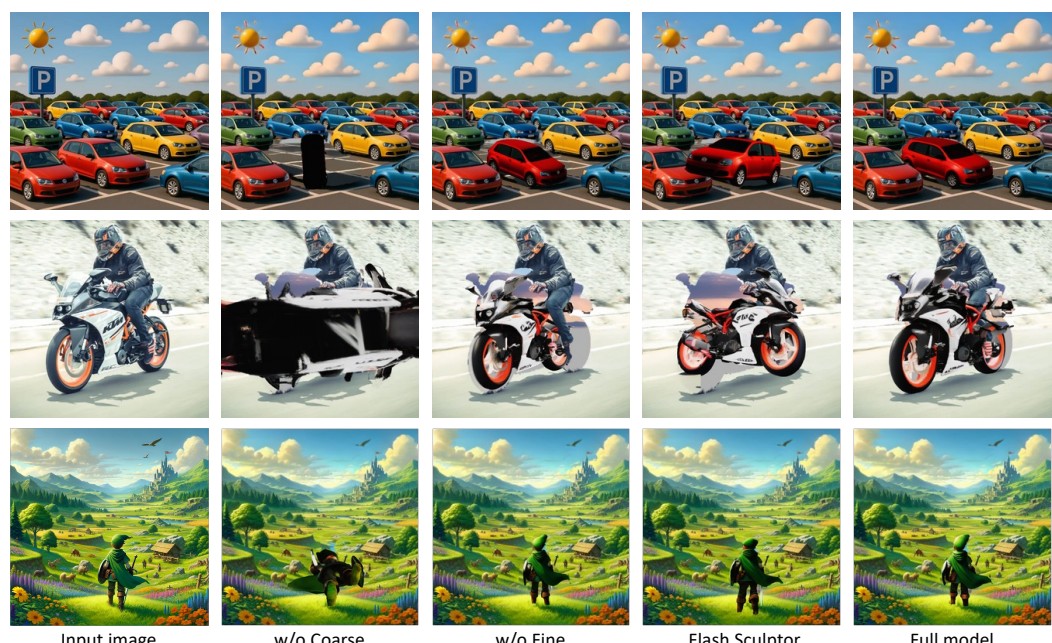

Figure 7: **Comparison of object alignment methods.**

Table 10: **The impact of in-context learning and prompt guidance on LLMs.** The results are derived from 8 diverse scenes, including both simple and complex cases.

| | Simulation | | Animation | | |
|---|---|---|---|---|---|
| Method | ObjAcc↑ | MotionAlign↑ | ObjAcc↑ | MotionAlign↑ | Penetration↓ |
| NeoWorld | **100** | **83.33** | **100** | **83.34** | **26.67** |
| w. in-context | **100** | 82.92 | **100** | 80.47 | 29.04 |
| w/o guidance | 87.5 | 28.33 | **100** | 70.95 | 33.33 |

As shown in Table. 10, the results show that our guidance achieves performance comparable to in-context learning, while removing all guidance leads to significant degradation, especially in simulation tasks, where outputs exhibit unrealistic physical parameters and incorrect material generation.

Furthermore, we find that NeoWorld outputs with and without in-context learning are often similar across many scenarios. This indicates that our guidance effectively fulfills the role of in-context learning by providing the model with essential cues to generate context-aware, high-quality results. It enhances the system's understanding of task requirements and helps infer correct attributes, similar to how structured examples guide in-context learning.

**Analysis of LLMs behaviors.** We analyze how the system handles ambiguous instructions by exposing its internal reasoning process and evaluating its robustness. For instance, given the prompt **"Move a boat slightly to the right"** in a scene with four boats and no further specification, the LLM generates the following internal reasoning trace:

> *The user wants to move a boat to the right. First, I need to figure out \*which\* boat to move. The scene has a few options, I'll review their positions and sizes to select a visually prominent one – a good starting point is one that's closer and larger. Instance ID 2 seems ideal; it's at the front (z=0.00) and has a reasonable size. So, Boat 2 it is.*
> *Next, I have to consider interactions, but I'm instructed to ignore background elements like the sea. So, the movement of the boat won't explicitly interact with the water, the simulation will only apply to the boat.*
> *Now for the material properties. The instructions specify 'jelly' with a high Young's modulus for rigid objects. I'll set a high Young's modulus, let's go with '1e6', and*

Table 11: **Ablation study of the fallback strategy in NeoWorld.**

| Method | CIQA+ | Q-Align | ImageCLIP | CS |
|---|---|---|---|---|
| w/o fallback | 0.58 | 2.68 | 77.81 | 72.03 |
| Full model | 0.59 | 2.66 | 78.63 | 72.46 |

Table 12: **Comparison of object removal methods**. We evaluate the removal model in terms of visual quality and unintended object emergence.

| Method | CIQA+ | Q-Align | Emergence rate↓ |
|---|---|---|---|
| LaMa | 0.71 | 4.28 | 17.85 |
| SD-2 inpainting | 0.72 | 4.32 | 37.04 |
| SDXL inpainting | 0.71 | 4.24 | 32.14 |
| SDXL removal | 0.71 | 4.30 | 7.40 |

*use a typical Poisson's ratio of '0.3'. For the friction angle, a standard '35' degrees will do. I'll go with a density of '800' kg/m³, a reasonable value for something like wood or composite.*
*Finally, I need to determine the force. The task is to move the boat slightly to the right, which is the positive x-direction. The magnitude of the force should be between 5 and 20, I'll set the force vector to '[10, 0, 0]'.*
*I'm now ready to formulate the JSON.*

This example illustrates how the LLM resolves ambiguity by leveraging spatial context, salience, and domain priors. It shows that the system consistently produces plausible and physically grounded decisions, even under incomplete instructions.

We also analyze how the system interprets complex relational prompts by exposing the LLM's internal reasoning process. Specifically, we consider the instruction: **"Move the right chair next to the left chair."**

*The user wants the right chair moved next to the left chair. My first task is to zero in on what's actually supposed to \*move\*. I'll need to locate the "right chair" and generate a trajectory for it. I confirm that instance ID 4 is indeed the "right chair" based on its X-coordinate being further right than the "left chair" at instance ID 6. Now for the animation. The right chair (ID 4) needs to end up beside the left chair. "Next to" implies a close proximity, and side-by-side is a good interpretation here. I calculate the final position of the right chair, so that its right edge is just touching the left chair's left edge. I will calculate the total delta for the right chair: I'll move it to the left, so it is next to the left chair, and also move it slightly back into the scene, to better align the two chairs. No y-axis movement. The math says: move it by -0.09 in the x direction, and +0.09 in the z direction.*
*Next, I need to generate a smooth animation path. I will use 100 timesteps. I'll use a smooth curve for the translation, based on a sine function to give a natural feel. The chair will slide over the floor. And for the rotation? For simplicity's sake, I'll go with no rotation at all throughout.*
*Finally, I'll generate the 100 keyframes for translation and rotation, generating the appropriate JSON format that will be passed to the user.*

This example illustrates the LLM's ability to handle complex instructions, including understanding intricate spatial relationships and avoiding collisions.

**Ablation study of object removal.** Unlike WonderWorld (Yu et al., 2025), which employs an inpainting model (Rombach et al., 2022) to remove foreground objects, we distill StableDiffusion-XL (Podell et al., 2024) into an 8-step student model specialized for this task using DMD2 (Yin et al., 2024). In Table. 12 and Fig. 16, we compare conventional inpainting models with our distilled model in terms of visual quality and unintended object emergence. These results show that while these methods produce comparable visual quality, the SDXL removal method significantly reduces semantic artifacts, which is critical for maintaining controllable and coherent scene editing.

Table 13: **The impact of the number of objects on manipulation accuracy.** We evaluate object selection accuracy and motion alignment.

| Num obj | ObjAcc↑ | MotionAlign↑ |
|---------|---------|--------------|
| 1 | 100 | 90.60 |
| 2 | 100 | 95.75 |
| 3 | 100 | 88.64 |
| 4 | 100 | 87.85 |

Table 14: **The model choice of LLMs.** The results are derived from 8 diverse scenes, including both simple and complex cases.

| Model | Simulation | | Animation | | |
| | ObjAcc↑ | MotionAlign↑ | ObjAcc↑ | MotionAlign↑ | Penetration↓ |
|-------|---------|--------------|---------|--------------|--------------|
| Qwen3-8B | **100** | 66.34 | **100** | 71.15 | 39.83 |
| Qwen3-30B-A3B | **100** | 61.06 | **100** | 62.09 | **28.85** |
| Gemini2.5Pro | **100** | **73.79** | **100** | **75.55** | 42.03 |

**Ablation study of fallback strategy.** In Fig. 17, we analyze the effect of fallback strategy in NeoWorld. The results show that the fallback strategy successfully filters failure cases arising from severe occlusions (1st row) and segmentation failures (2nd row). In Table. 11, we further quantify this effect: the differences with and without fallback are marginal, indicating that such failures are infrequent and underscoring the overall robustness of NeoWorld.

**Multi-object manipulation.** In Table 13, we analyze the impact of the number of objects on manipulation performance, measured by object selection accuracy and motion alignment. For each number of objects, we evaluate on 6 different cases. The results indicate that, as the number of objects increases, NeoWorld consistently selects all target objects correctly, and motion alignment remains largely unaffected, demonstrating the effectiveness and robustness of our object-centric representation.

**LLM choices.** In Table 14, we further evaluate different LLM choices, including Gemini2.5Pro used in NeoWorld, as well as two open-source lightweight models: Qwen3-8B-Thinking and Qwen3-30B-A3B-Thinking-2507 (Yang et al., 2025). The results show that all models can reliably select the correct target objects and achieve similar performance, highlighting the effectiveness of the object-centric scene representation and control interface in NeoWorld.

**Scalability analyses.** In Table 15, we report generation time, peak GPU memory, and storage of the generated world as we increase the world size and the number of unfolding objects, where world size refers to the number of scenes contained in the generated world. These trends indicate that our system scales approximately linearly in time and storage with respect to both world size and the number of unfolding objects, while incurring minimal additional GPU memory overhead.

## C  DETAILED QUANTITIVE RESULTS

The benchmark of NeoWorld includes 7 distinct styles and occlusion conditions:

- **Photorealistic**: Realistic environments with detailed textures and geometry.
- **Ink Painting**: Highly abstract visuals featuring brush-like textures.
- **Oil Painting**: Scenes with rich, layered colors and blended geometric edges.
- **Cyber-punk**: Futuristic, neon-lit environments with dense layouts and visual clutter.
- **Minecraft**: Blocky, pixelated worlds with low-resolution textures.
- **Anime**: Stylized 2D visuals with vibrant palettes and simplified geometric representations.
- **Complex Scenes**: High object occlusions and intricate layouts.

Table 15: **Scalability with respect to world size and the number of unfolding objects.** We report generation time, peak GPU memory footprint, and storage for the generated world.

| Scalability | | Time(s) | Memory footprint | Storage |
|---|---|---|---|---|
| World size | 1 | 18.14 | 23.29G | 236M |
| | 2 | 37.19 | 24.20G | 259M |
| | 4 | 73.15 | 24.23G | 301M |
| | 8 | 150.13 | 24.38G | 373M |
| | 16 | 293.98 | 24.84G | 524M |
| Unfolding objects | 1 | 18.14 | 23.29G | 236M |
| | 2 | 25.87 | 23.29G | 284M |
| | 3 | 33.02 | 23.30G | 355M |

Table 16: **Performance on different types of scenes for interactive world generation.**

| | Photorealistic | | | | Ink painting | | | |
|---|---|---|---|---|---|---|---|---|
| Method | Q-Align | Clip-Score | 3D-Const | SceneQuality | Q-Align | Clip-Score | 3D-Const | SceneQuality |
| WonderJourney | 1.71 | 59.05 | 18.45 | 18.19 | 1.53 | 63.03 | 22.86 | 23.81 |
| WonderWorld | 2.45 | 67.32 | 39.31 | 34.38 | 1.90 | 62.85 | 28.57 | 28.57 |
| NeoWorld | **2.84** | **69.78** | **42.24** | **47.43** | **2.33** | **66.16** | **48.57** | **47.62** |

| | Oil painting | | | | Cyber-punk | | | |
|---|---|---|---|---|---|---|---|---|
| Method | Q-Align | Clip-Score | 3D-Const | SceneQuality | Q-Align | Clip-Score | 3D-Const | SceneQuality |
| WonderJourney | 1.67 | 68.38 | 14.29 | 20.00 | 1.56 | 72.00 | 25.24 | 21.90 |
| WonderWorld | **2.95** | 63.16 | 31.43 | 29.52 | 2.16 | 72.13 | 28.57 | 29.06 |
| NeoWorld | **2.95** | **64.86** | **54.29** | **50.48** | **2.37** | **74.94** | **46.19** | **49.04** |

In Tables 16-17, we present the detailed performance of NeoWorld across different scene categories. The results show that NeoWorld consistently surpasses the baseline models and demonstrates robustness across diverse image styles, including challenging cases with occlusions and visual clutter.

# D MORE VISUALIZATION RESULTS

Fig. 8-10 compare the exploration and novel views generated by different methods. In Fig. 11, we also present interactive world generation results using exploration views only. The 2D video diffusion models (*e.g.*, Wan2.1-I2V) lack explicit control over camera trajectories and tend to produce frames that closely resemble the input image. The 2D interactive method Matrix-Game2 fails to provide accurate camera control and does not preserve object-level 3D consistency. Furthermore, compared to existing interactive world generation methods such as WonderWorld and WonderJourney, which rely on surface-level representations, though WonderWorld and NeoWorld achieve comparable visual quality in exploration views, our method demonstrates significantly higher 3D consistency in the generated views. In Fig. 12, we also include visualizations of dynamic scene simulations annotated with user prompts, illustrating how our method responds to motion-specific instructions and maintains temporal coherence across frames.

In Fig. 13, we present visual results of dynamic scene simulation and animation involving multiple objects, demonstrating the effectiveness and robustness of our object-centric representation in handling complex multi-object interactions.

In Fig. 15, we further showcase the visualizations of translation, rotation, and animation. For the animation, the 3D character is reconstructed with an existing Image-to-3D tool (Tripo 3D (Tripo 3D, 2025)) and subsequently animated using Mixamo (Adobe Inc., 2025).

Additionally, because all manipulations are performed directly in 3D and then rendered, our method can generate images from arbitrary viewpoints and time steps. In Fig. 14, we present the same dynamic scenes rendered from two static cameras (1st–2nd rows) and two moving cameras (3rd–4th rows).

Table 17: **Performance of interactive world generation (Part 2).** Metric names are abbreviated for compact presentation.

| Method | MineCraft | | | | Anime | | | | Complex | | | |
|---|---|---|---|---|---|---|---|---|---|---|---|---|
| | QA | CS | 3DCons | SQ | QA | CS | 3DCons | SQ | QA | CS | 3DCons | SQ |
| WonderJourney | 1.69 | 73.93 | 19.05 | 22.86 | 1.79 | 64.59 | 17.38 | 16.19 | 2.02 | 74.41 | 25.40 | 25.71 |
| WonderWorld | 2.39 | 79.27 | 33.33 | 29.52 | 2.03 | 69.53 | 23.33 | 32.62 | 2.39 | 72.04 | 29.84 | 33.33 |
| NeoWorld | **2.45** | **81.42** | **47.62** | **47.62** | **2.69** | **72.12** | **59.29** | **51.19** | **2.65** | **75.86** | **44.76** | **40.96** |

Table 18: **Per-module runtime (in seconds).** "All" denotes the end-to-end runtime of the full pipeline over a single scene.

| Image inpainting | Depth estimation | Object removal | Segmentation |
|---|---|---|---|
| 2.08 | 1.70 | 1.50 | 0.51 |
| Gaussian training | 3D unfolding | Alignment | All |
| 5.01 | 5.42 | 1.92 | 18.14 |

# E FURTHER IMPLEMENTATION DETAILS

## E.1 GAUSSIAN LAYER INITIALIZATION

Following WonderWorld (Yu et al., 2025), we adopt guided depth diffusion using marigold depth and marigold normals to initialize the geometry of Gaussian layers. Specifically, given a scene image $I_i$, the guided depth diffusion estimates the depth based on existing geometries (i.e., the depth rendered from previously constructed scenes), ensuring multi-scene geometric coherence. Next, normals are computed using Marigold normals.

Each pixel is then initialized as a 2D Gaussian, where the position is derived from its pixel coordinate and depth, the quaternion is computed from the normals, the color is set based on the corresponding pixel color, and the scale is determined according to the Nyquist sampling theorem. During optimization, the position and color remain fixed, while the scale, opacity, and quaternions are updated to refine the representation.

## E.2 PER-MODULE TIME BREAKDOWN

In Table 18, we report a per-module runtime breakdown of our pipeline, providing a more detailed characterization of its overall efficiency. This breakdown indicates that the main computational bottlenecks lie in the 3D stages of our pipeline(including Gaussian layers training and 3D unfolding), while the cost of 2D modules is already modest.

## E.3 HUMAN STUDY DETAILS

We recruited 105 participants for a blind preference study. In each trial, participants were shown video clips generated by different methods for the same scene. The method order is randomized per trial. Participants are instructed to select exactly one best video based on 3D consistency, scene quality, and other metrics. The survey is fully anonymous. We report results as preference rates, i.e., the percentage of trials in which each method is chosen.

## E.4 LLM-BASED USER INTERACTION

In user interaction and dynamic simulation, we employ an LLM $g_{\text{LLM}}$ to derive the target object index and manipulation attributes: $\mathcal{I}, \mathcal{A} = g_{\text{LLM}}(\mathcal{J}, \mathcal{O}, \mathcal{U})$. where $\mathcal{J}$ represents the instruction prompt, $\mathcal{O}$ contains the object-related information, and $\mathcal{U}$ denotes the user input prompt. Specifically, object-related information $\mathcal{O}$ comprises the 3D position and size of each object, as well as its instance index and category.

For the simulation task, the instruction prompt $\mathcal{J}$ describes the intended dynamics of the scene. For the animation task, a similar instruction prompt is used; however, the output is extended to include a sequence of translations and rotations applied to each object instance, enabling fine-grained control over individual motions.

The instruction prompt $\mathcal{J}$ for the simulation task is defined as follows:

*You are a simulation assistant. Next, you will be provided with object information in a scene and a user prompt. You need to identify the foreground objects most likely to interact with each other, and estimate appropriate material point method (MPM) attributes for each. When selecting an object to simulate:*

1. *Pay close attention to any spatial indicators in the user prompt (e.g., "the apple on the left", "the top plate", "the apple falling onto the plate").*
2. *Consider object descriptions (e.g., position, size) when multiple objects of the same category exist.*
3. *Select objects that are mentioned in the user prompt or are likely to participate in the described interaction.*
4. *Most scenes involve 1–3 foreground objects interacting with each other.*
5. ***Coordinate system:** Defined as follows: $+x$ points to the right of the image, $+y$ points upward, and $+z$ points into the scene (i.e., away from the viewer).*

*For each selected object, you should provide simulation parameters including:*

- ***Material type:** Choose from the following list: `['jelly', 'sand', 'foam', 'snow', 'plasticine']`.*
- ***Young's modulus (E):** Represents stiffness. Higher values indicate stiffer materials.*
- ***Poisson's ratio (nu):** Represents how much a material contracts in directions perpendicular to the direction it is stretched.*
- ***Density** and **Friction angle** should be set appropriately based on the material and object type.*
- ***Force:** Provide a 3D vector $[f_x, f_y, f_z]$ representing the applied force, which should be set appropriately based on the description of dynamics in the user prompt. Suitable force magnitudes typically range from 5 to 20 to create visible motion and interaction effects.*

*Here's a guide to help you select the appropriate material:*

- *jelly: For elastic objects that can deform and return to their original shape (like rubber, soft fruits, gelatin-like substances). Best for simulating bouncy, elastic objects. Young's modulus (E): 1e4-1e6, Poisson's ratio (nu): 0.3-0.45*
- *sand: For granular materials that can flow but maintain volume (like sand, sugar, rice). Best for simulating grainy substances that pour. Young's modulus (E): 1e6-1e8, Poisson's ratio (nu): 0.2-0.3, $friction_angle : 30 - 45$*
- *foam: For soft, compressible materials that absorb impact (like cushions, sponges, styrofoam). Young's modulus (E): 1e3-1e5, Poisson's ratio (nu): 0.1-0.3*
- *snow: For brittle, lightweight materials that can break apart and accumulate (like snow, powder). Young's modulus (E): 1e4-1e6, Poisson's ratio (nu): 0.2-0.3*
- *plasticine: For materials that deform permanently and don't return to original shape (like clay, dough, plasticine). Best for simulating objects that can be molded. Young's modulus (E): 1e5-1e7, Poisson's ratio (nu): 0.3-0.4*

*For rigid objects like furniture, use 'jelly' with a high Young's modulus (E: 1e5-1e7). For soft objects like fruits, pillows, use 'jelly' with low Young's modulus (E: 1e2-1e4). For moldable objects like clay or dough, use 'plasticine'. For grainy substances like sugar or salt, use 'sand'.* Please use the following JSON format for the output:

```
{
  "objects": [
    {
      "instance_id": instance_id_1,
      "material_params": {
        "material": material_1,
        "E": E_1,
        "nu": nu_1,
        "friction_angle": friction_angle_1,
        "density": density_1
      },
      "force": [f_x_1, f_y_1, f_z_1]
    },
    {
      "instance_id": instance_id_2,
      "material_params": {
        "material": material_2,
        "E": E_2,
        "nu": nu_2,
        "friction_angle": friction_angle_2,
        "density": density_2
      },
      "force": [f_x_2, f_y_2, f_z_2]
    }
  ]
}
```

Finally, we apply several lightweight post-processing steps to improve the quality of LLM outputs. For simulation, we clamp generated force values to a physically plausible range to ensure stable, realistic dynamics. For animation, we resample and interpolate translation and rotation trajectories to match the target duration, since the LLM outputs may not perfectly align with the intended length. We also apply a temporal smoothing filter to the translation and rotation signals to produce coherent, artifact-free motion.

## F    FAILURE CASE ANALYSIS

Despite incorporating a fallback strategy and several robustness mechanisms, failures can still occur under severe occlusions or segmentation errors. Fig. 18 illustrates typical cases: (i) alignment errors (1st row), where the reconstructed 3D object is misaligned with the target, yielding incoherent results; (ii) image-to-3D degradation (2nd row), where the image-to-3D module either fails to recover fine object details—leading to visual degradation—or lacks sufficient cues under heavy occlusion, causing failures; and (iii) segmentation errors (3rd row), where over- or under-segmentation produces inaccurate 3D geometry.

In Table 19, we report the empirical failure frequency (in %) of each module in our pipeline. Overall, the failure rates are low, and errors are primarily concentrated in the image-to-3D, alignment, and segmentation modules. Image-to-3D failures mostly occur when reconstructing humans or objects with highly complex geometry. Alignment failures typically arise in scenes with severe occlusions or highly cluttered object configurations. Since OneFormer is a closed-set panoptic segmenter, segmentation failures are mainly due to out-of-distribution categories. In contrast, failures from the depth estimator and the LLM are relatively rare. Taken together, these statistics demonstrate the robustness and effectiveness of NeoWorld.

In Fig. 19, we compare the Amodal3R used in NeoWorld with a very recent open-source model SAM3D (Team et al., 2025), as well as the closed-source model Tripo3D. We observe that these latest image-to-3D models already produce significantly better visual quality than earlier approaches. We expect NeoWorld to continue benefiting from future advances in image-to-3D, leading to increasingly faithful and detailed object reconstructions.

Table 19: **Per-module failure frequency (%) on our benchmark.**

| Depth | Image-to-3D | Alignment | Segmentation | LLM |
|-------|-------------|-----------|--------------|-----|
| 0.83 | 3.33 | 2.50 | 5.83 | 0.83 |

To address these limitations, promising directions include employing more capable image-to-3D models for both reconstruction and alignment, refining masks with interactive segmentation methods (*e.g.*, SAM (Kirillov et al., 2023)), and replacing the current fallback scheme with a multimodal large language model to further improve robustness.

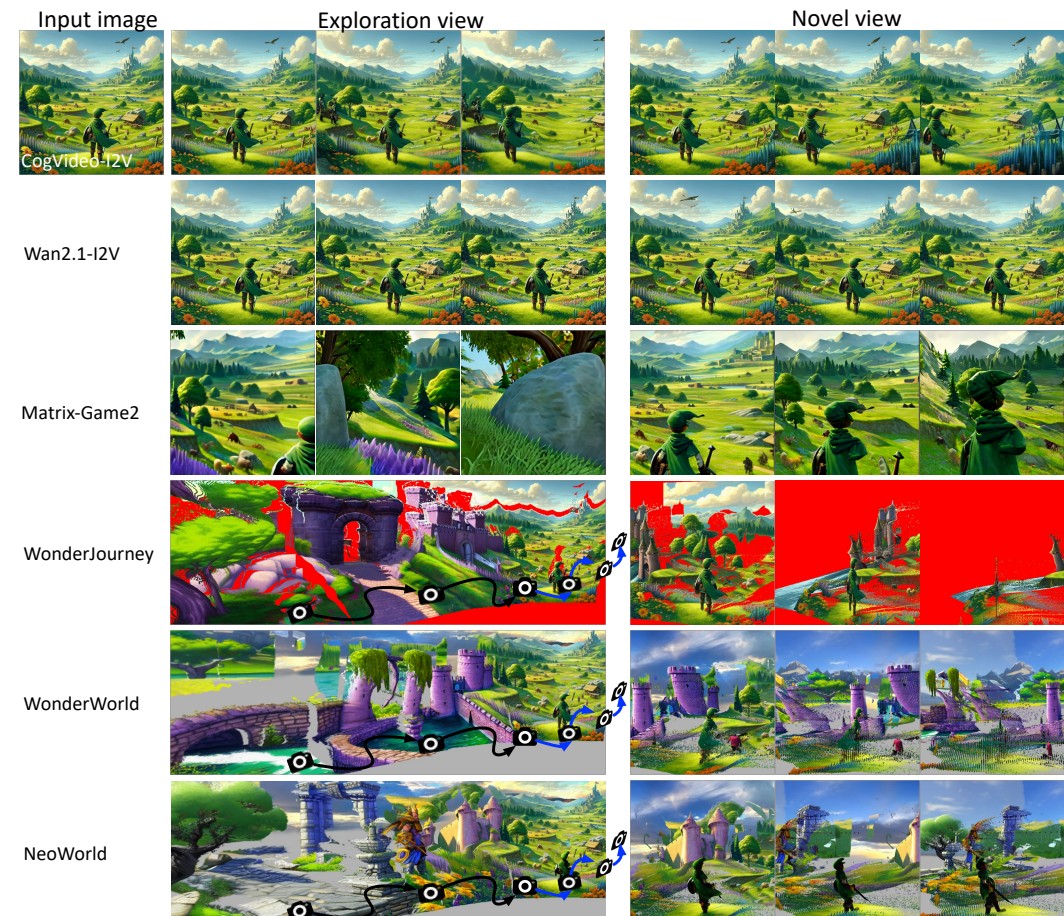

Figure 8: **Additional examples of interactive world generation (Part 1).**

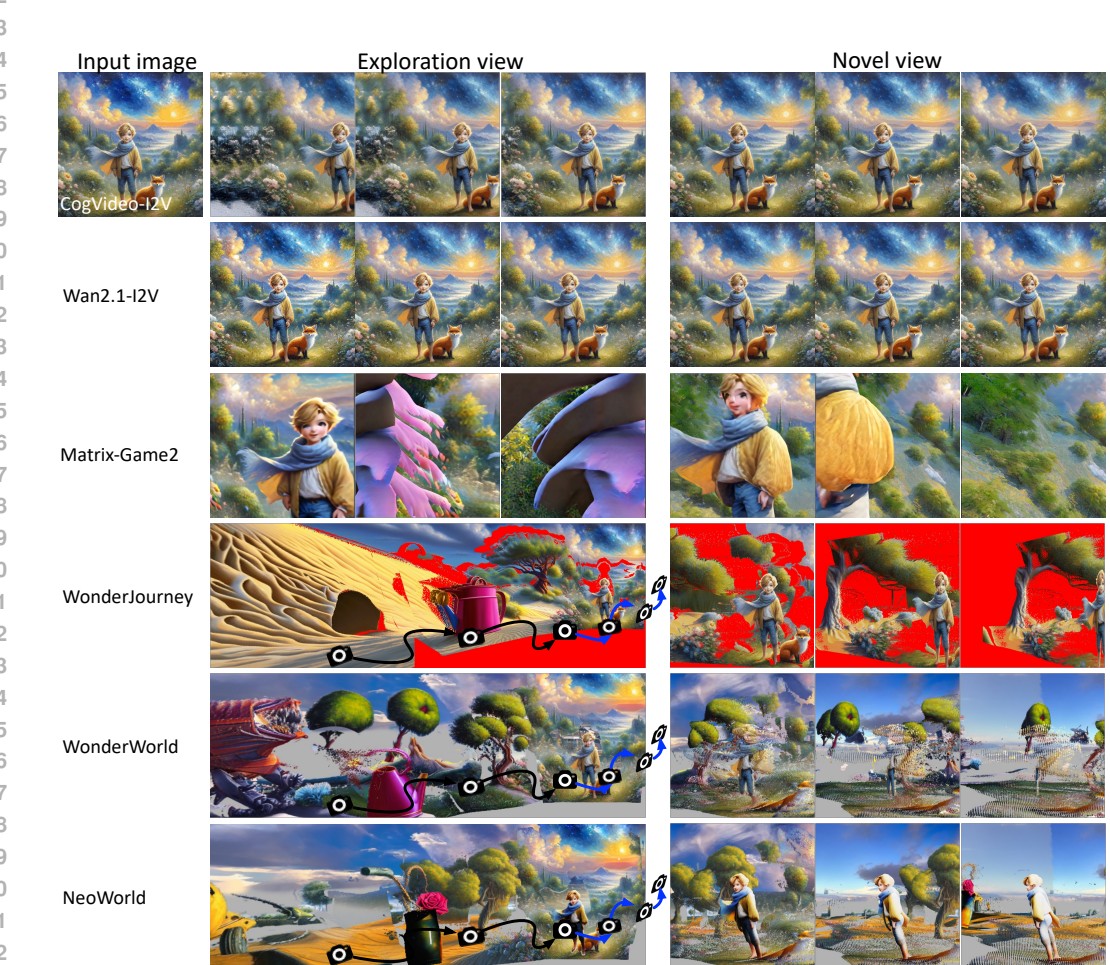

Figure 9: **Additional examples of interactive world generation (Part 2).**

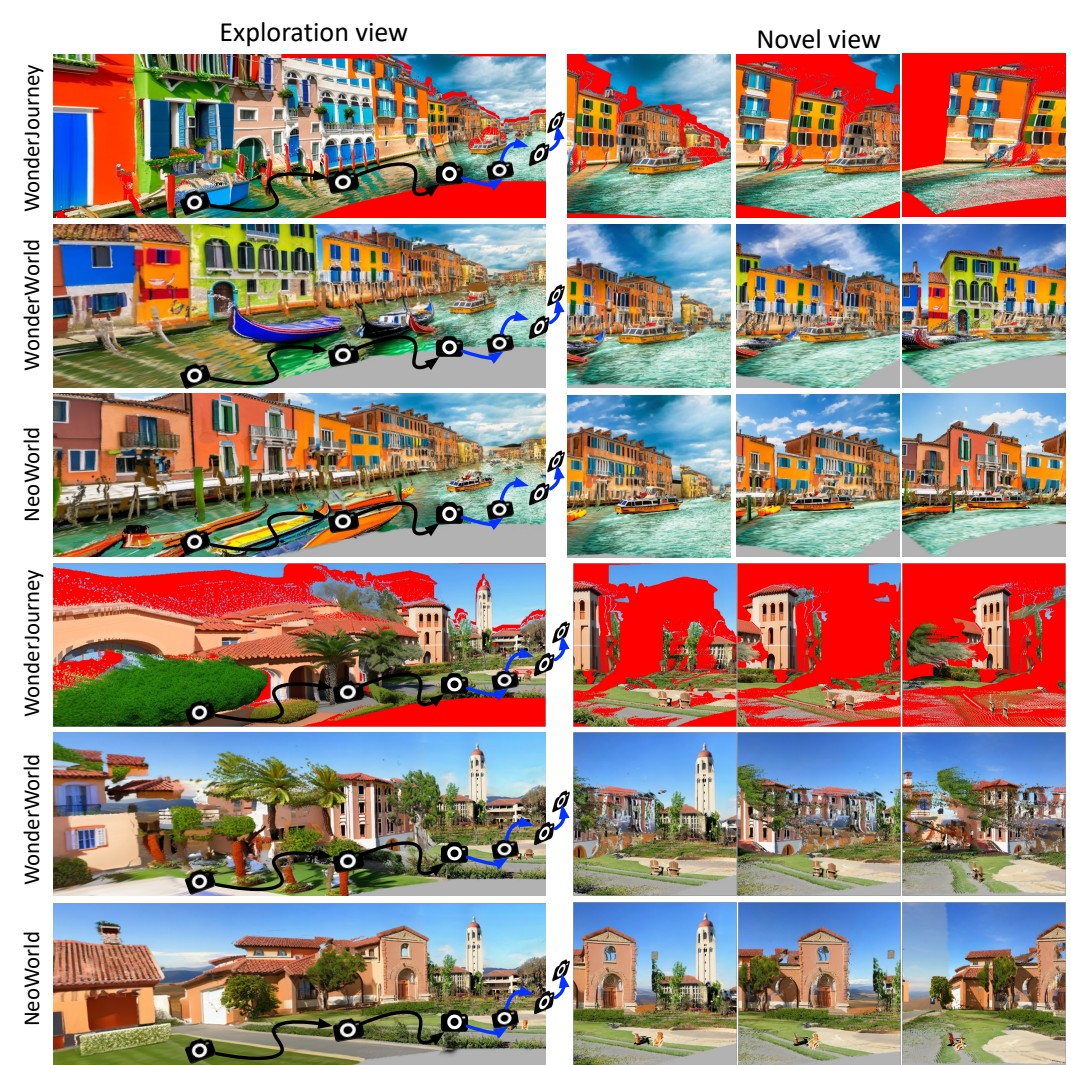

Figure 10: **Additional examples of interactive world generation (Part 3).**

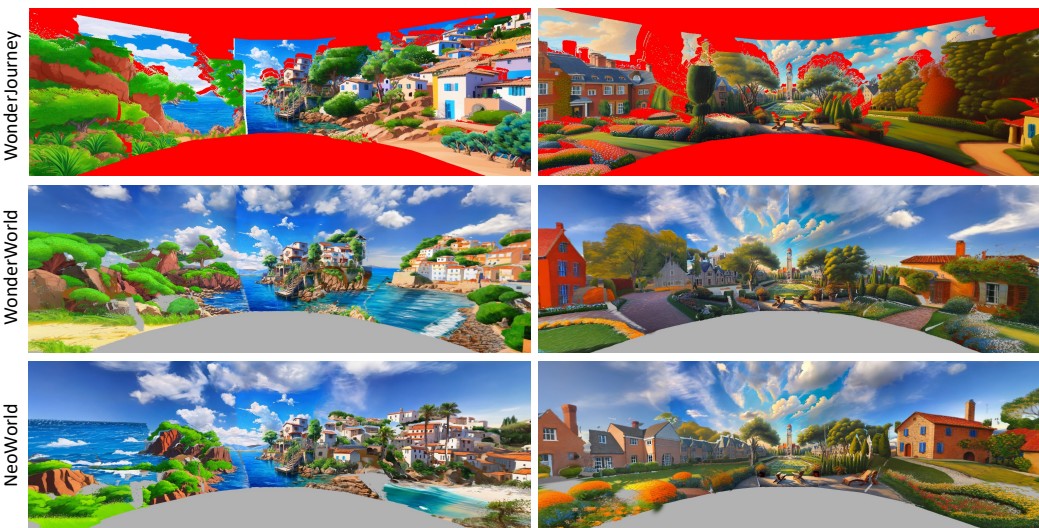

Figure 11: **Additional examples of interactive world generation with exploration views.**

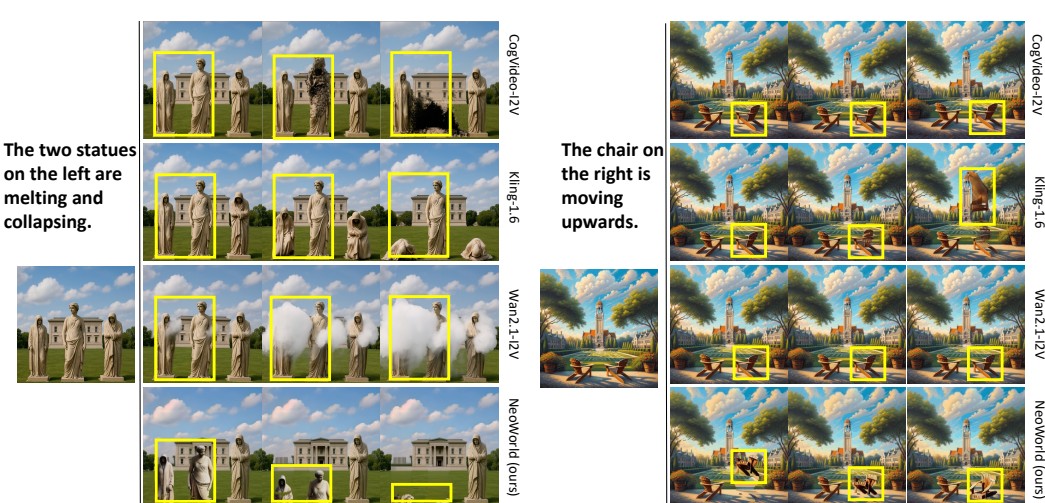

Figure 12: **Showcases of dynamic scene simulation.**

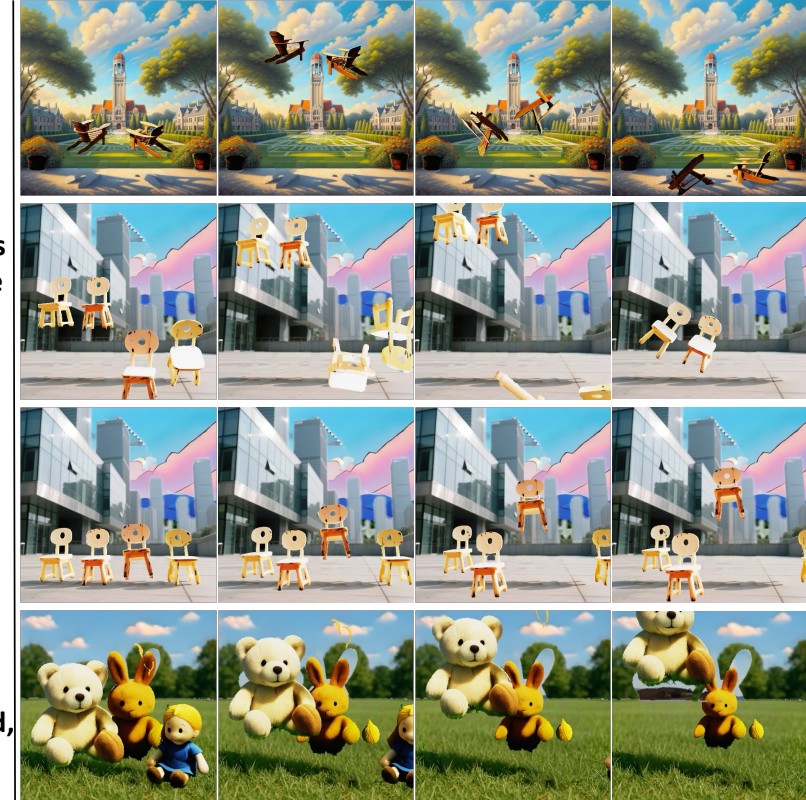

Figure 13: **Demonstration of multiple objects manipulation.** We show examples of dynamic scene simulation in 1st–2nd rows and manipulation 3rd–4th rows.

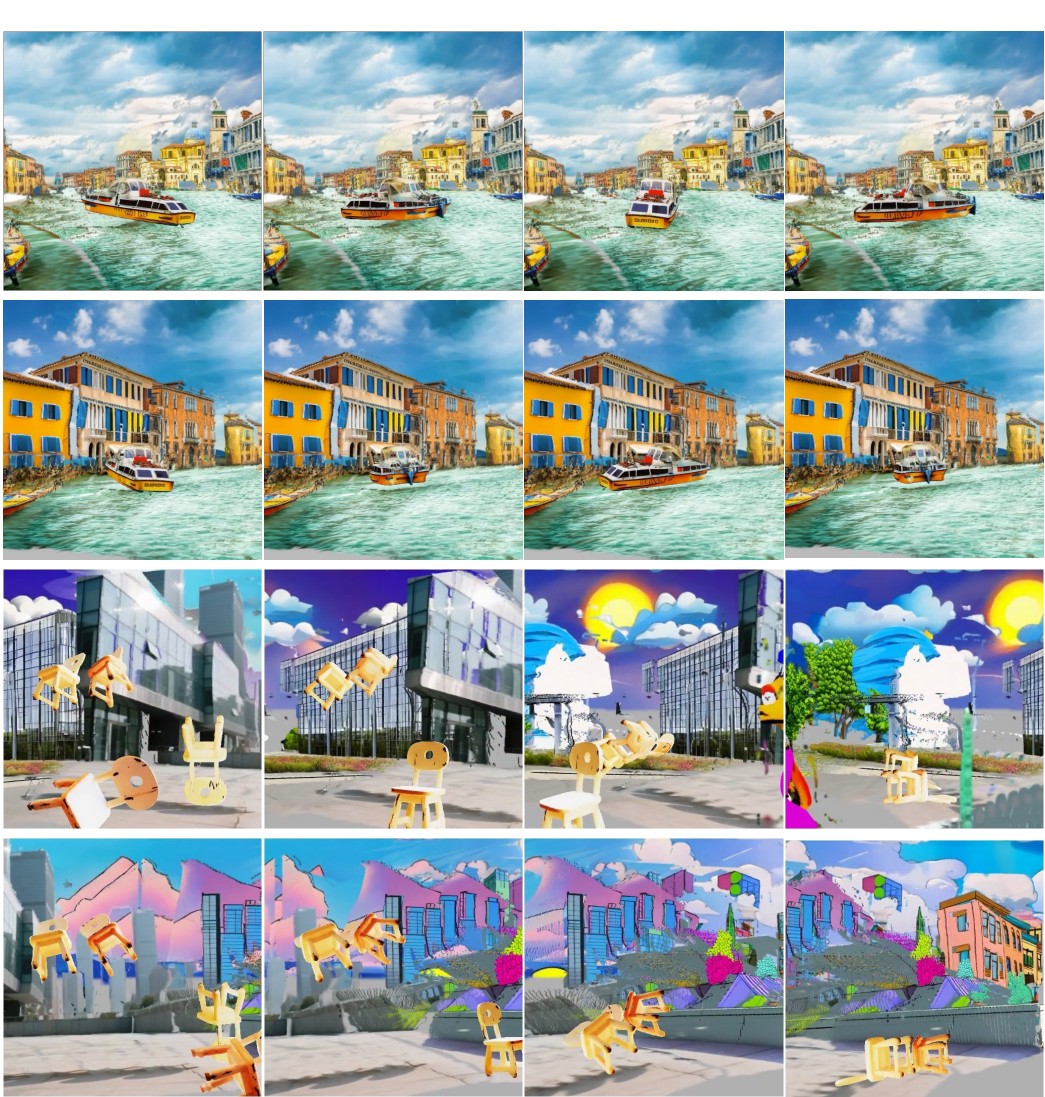

Figure 14: **Multi-view visualizations of dynamic scenes under different camera settings.** We render the same dynamic scenes from two static cameras (1st–2nd rows) and two moving cameras (3rd–4th rows).

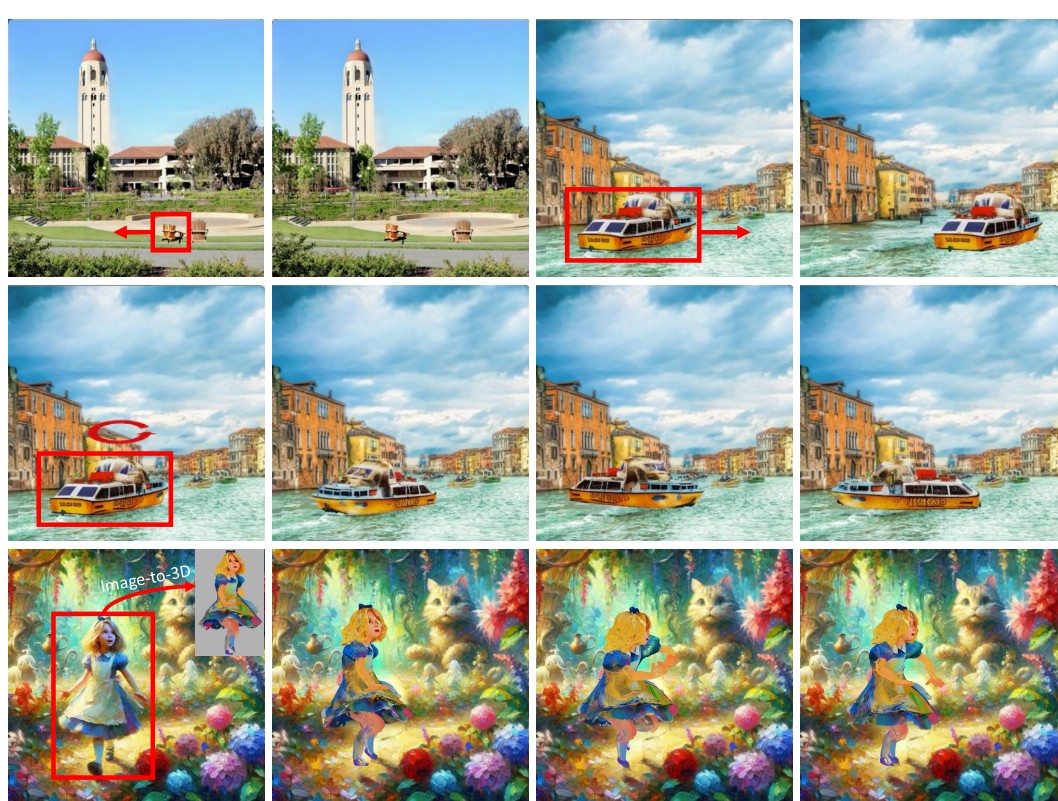

Figure 15: **Qualitative results of manipulation: transition (1st row), rotation (2nd row), and animation (3rd row).**

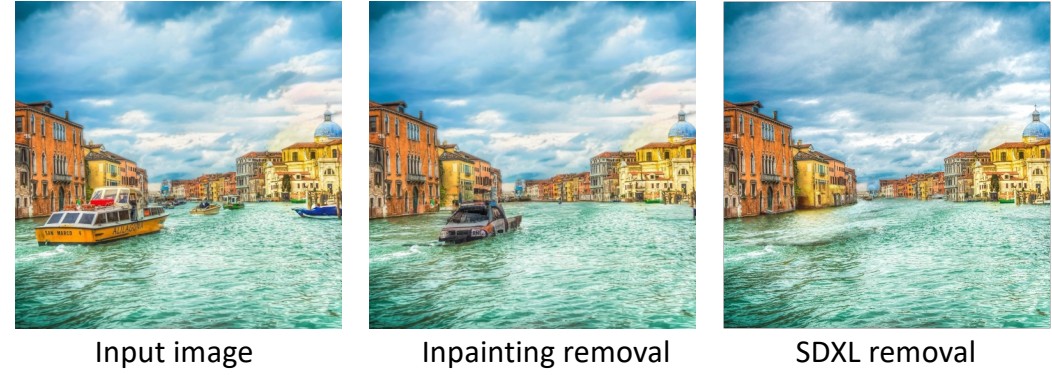

Input image                    Inpainting removal                    SDXL removal

Figure 16: **Comparison of object removal methods.** The inpainting-based removal result (middle) introduces unintended artifacts and objects, which can complicate subsequent scene generation. To address this issue, we adopt the distilled SDXL specialized for this task (right), which yields cleaner and more controllable removal results.

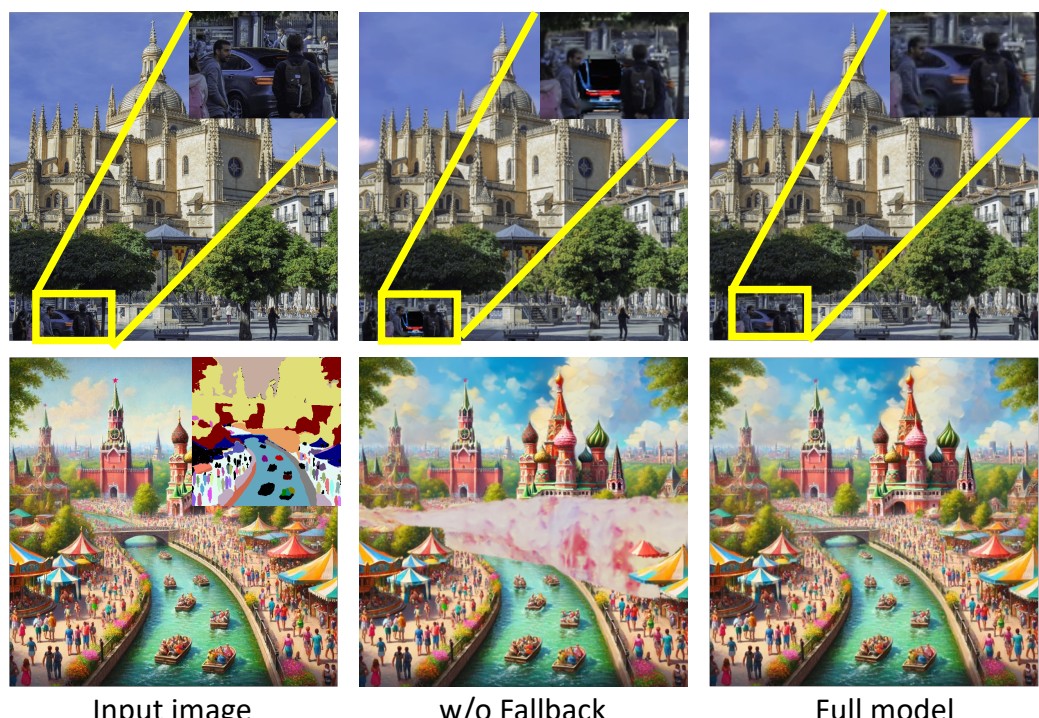

Input image      w/o Fallback      Full model

Figure 17: **Comparison of world unfolding results with and without fallback.** Fallback effectively filters out common failures caused by image-to-3D degradation (1st row) and segmentation errors (2nd row).

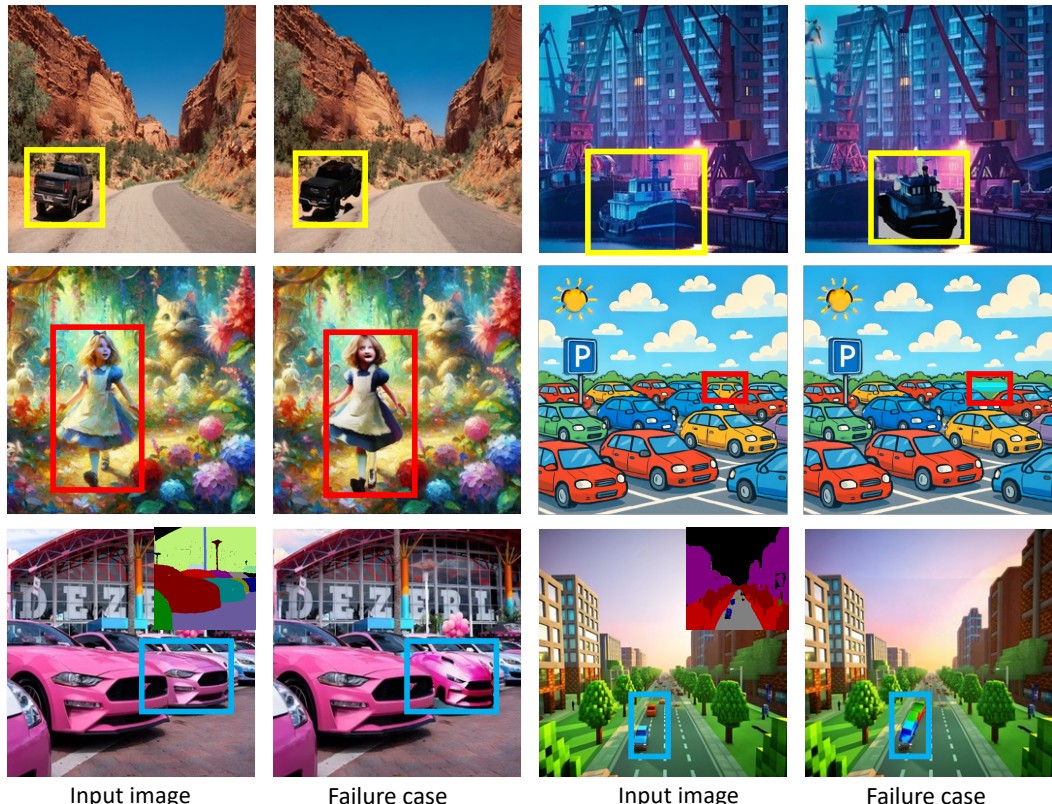

Input image     Failure case      Input image     Failure case

Figure 18: **Visualizations of failure cases.** Examples of failures caused by alignment (1st row), image-to-3D degradation (2nd row), and segmentation errors (3rd row).

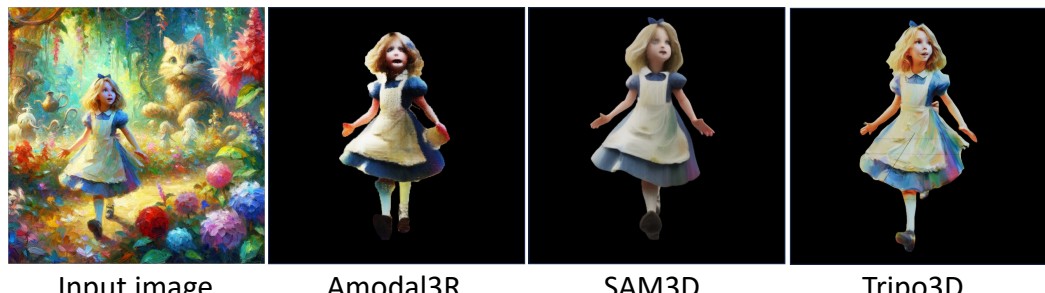

Input image          Amodal3R          SAM3D          Tripo3D

Figure 19: **Comparison of different image-to-3D backbones.**

