# OpenReview forum: "NeoWorld: Neural Simulation of Explorable Virtual Worlds via Progressive 3D Unfolding"
_ICLR.cc/2026/Conference — Submitted to ICLR 2026_

### Official Review · Reviewer_rktf · 2025-10-27

**Soundness:** 3
**Presentation:** 3
**Contribution:** 2
**Rating:** 4
**Confidence:** 3

**Summary:**

The paper proposed NeoWorld, a framework designed for generating interactive 3D virtual worlds from a single input image. It introduces a hybrid object-centric scene representation that progressively transforms a 2D scene into high-fidelity 3D models, allowing dynamic world exploration. The approach utilizes cutting-edge representation learning, object-centric techniques, and language models for flexible viewpoint manipulation and controlling object appearance or dynamics. The latter experiment demonstrate its computational efficiency with high-level realism.

**Strengths:**

1. The idea of modeling virtual worlds interactively, where only the explored regions are rendered in detail, is quite compelling. The paper's motivation of merging object-centric 3D rendering with computational efficiency is both interesting and valuable for creating immersive environments.
2. The system's ability to support language-driven object manipulation, along with physics-based animation. The demonstration of the object dynamic interactions (e.g., chair collision and boat bouncing) shows the potential of this system to create physically plausible simulations. This feature is particularly impressive for an interactive virtual world generation framework.
3. NeoWorld combines several advanced techniques, including object-centric neural scene representations, 2.5D-to-3D unfolding, and user–scene interaction module. These innovations empower the interactive world generation.

**Weaknesses:**

1. Although the paper mentions the ability for users to manipulate objects within the scene, the example provided in the paper is more like edited-related cases; it is unclear if the framework could achieve dynamic object manipulation while the camera navigates with a trajectory. If it only supports object manipulation in a static camera pose, the contribution might be overclaimed.
2. The author provides only a single demonstration of multi-object manipulation case in the appendix (two statues melting), the performance degradation analysis when the number of objects increases will be helpful to support the "object-centric" contribution.

**Questions:**

Refer to my weakness paragraph.

---

> ### Author Response · Authors · 2025-11-24
>
> We appreciate the reviewer’s constructive comments. Please refer to our revised paper for the newly added results and corresponding visualizations.
>
> > Q1. Although the paper mentions the ability for users to manipulate objects within the scene, the example provided in the paper is more like edited-related cases; it is unclear if the framework could achieve dynamic object manipulation while the camera navigates with a trajectory. If it only supports object manipulation in a static camera pose, the contribution might be overclaimed.
>
> As shown in **Fig. 14** in the revision, our framework supports **simultaneous object motion and camera motion**, not just editing under a static viewpoint. Because all manipulations are performed directly in 3D before rendering, we can generate images from arbitrary viewpoints and timestamps. This enables scenarios such as:
> - Moving objects with a moving camera following a trajectory,
> - Moving objects observed by multiple static cameras, and
> - Moving objects observed by multiple moving cameras.
>
> All these cases preserve 3D consistency across views and time, which is precisely what 2D video generation models cannot reliably guarantee. In Fig. 14, we illustrate this by rendering the **dynamic scenes from two moving cameras (3rd–4th rows)**.
>
>
>
> > Q2. The author provides only a single demonstration of a multi-object manipulation case in the appendix (two statues melting), the performance degradation analysis when the number of objects increases will be helpful to support the "object-centric" contribution.
>
> Thank you for the suggestion. In this revision, we have added more comprehensive multi-object manipulation results, both quantitatively (**Table 13**) and qualitatively (**Fig. 13**). The results show that as the number of objects increases, NeoWorld consistently selects all target objects correctly, and motion alignment remains largely unaffected, demonstrating the effectiveness and robustness of our object-centric representation.

---

### Official Review · Reviewer_kLxv · 2025-10-27

**Soundness:** 2
**Presentation:** 3
**Contribution:** 2
**Rating:** 6
**Confidence:** 3

**Summary:**

This work presents NeoWorld, a system for interactive 3D world generation from a single image. It progressively expands 3D scenes during user exploration, combining efficiency with realism by using a hybrid structure—2D backgrounds and fully 3D foreground objects. NeoWorld integrates differentiable rendering and image-to-3D reconstruction to enable 3D-consistent exploration and physics-based interaction, surpassing prior 2D and 2.5D methods.

**Strengths:**

1. Comparing to prior works, it now support interaction with foreground objects.
2. It doesn't sacrifice too much time cost for bringing only the foreground objects to 3D.
3. It outperform prior works in terms of PromptAlign and some other criteria.

**Weaknesses:**

1. Some distortions remain visible in the qualitative results.
2. Object movement is still limited to basic, language-driven control. In realistic settings, objects move only through physical interactions, which may explain why current video generation models struggle with such behavior.
3. The reconstruction quality of the object is low. Is there any way to improve that?

**Questions:**

Could the generated motions could be more complex and realistic? For instance, the car in Figure 3 could move naturally along the road.

---

> ### Author Response · Authors · 2025-11-24
>
> We appreciate the reviewer’s insightful questions and provide our responses below. Please refer to our revised paper for the newly added results and corresponding visualizations.
>
> > Q1. Some distortions remain visible in the qualitative results.
>
> The remaining distortions in novel views mainly arise from background regions (e.g., trees, buildings) that are modeled in 2.5D. As discussed in the manuscript, full-scene 3D generation can largely eliminate such artifacts but is computationally prohibitive in our interactive setting.
>
> In practice, noticeable distortions appear only under large viewpoint shifts in the background regions. This motivates our hybrid scene representation: foreground objects—where user interaction is frequent and novel-view artifacts are more critical—are modeled in full 3D to maximize visual quality, while the background remains in 2.5D to keep the system efficient. Finally, by applying the V2V model described in Sec. 3.4, we can further enhance visual quality and substantially reduce the remaining distortions.
>
> >Q2. Object movement is still limited to basic, language-driven control. In realistic settings, objects move only through physical interactions, which may explain why current video generation models struggle with such behavior.
>
> We agree that accurately modeling physical interactions between objects remains a major open challenge for current video generation systems. Our work does not completely solve this problem. Nonetheless, rather than relying purely on language-driven motion, our framework takes an important step forward by introducing an explicit 3D, MPM-driven, object-centric representation within an interactive, generated world. This representation is naturally compatible with more advanced physics engines such as PhysX or Bullet. We view this as a promising direction for future work, in which our current MPM-driven control can be further extended to support more realistic, physically grounded object motion and interactions.
>
>
> > Q3. The reconstruction quality of the object is low. Is there any way to improve that?
>
> Yes. Current open-source methods (e.g., Trellis, Amodal3R) are constrained by their training datasets and tend to perform poorly on complex objects or humans. In our setting, one way to improve visual quality is to apply the visual enhancement module introduced in our revision: after obtaining the dynamic video, we use a video-to-video (V2V) model to refine appearance and enhance perceptual quality.
>
> In parallel, as image-to-3D methods continue to advance, the underlying reconstruction quality will steadily improve. For example, in Fig. 19 we showcase results from the very recent SAM3D and a closed-source model, Tripo3D, both of which already achieve significantly better visual quality than earlier approaches. We expect our framework to continue benefiting from such progress in image-to-3D, leading to increasingly faithful and detailed object reconstructions.
>
>
> > Q4. Could the generated motions be more complex and realistic? For instance, the car in Figure 3 could move naturally along the road.
>
> Yes. By using prompts such as “The car is driving towards us” or “The boat is moving away”, NeoWorld can generate more complex and realistic motions, including cars moving naturally along the road. On top of this, our visual enhancement module further improves realism and scene coherence, producing natural lighting and shadow variations as well as realistic water dynamics (see **Fig. 6**). We refer the reviewer to the **MP4** files in the supplementary materials for more comprehensive qualitative results.

---

### Official Review · Reviewer_UroN · 2025-10-29

**Soundness:** 3
**Presentation:** 2
**Contribution:** 2
**Rating:** 4
**Confidence:** 3

**Summary:**

This paper, NeoWorld: Neural Simulation of Explorable Virtual Worlds via Progressive 3D Unfolding, presents a system that builds interactive, explorable 3D worlds from a single input image. The method begins with an object-centric 2.5D Gaussian representation (with collapsed depth per primitive), progressively converting selected regions into full 3D reconstructions as the user navigates or issues natural-language commands. The system integrates several pretrained components — panoptic segmentation (OneFormer), depth/normal estimation (Marigold), image inpainting (Stable Diffusion v2.0 / distilled SD-XL), image-to-3D reconstruction (Amodal3R), and an LLM (Gemini-2.5pro) for object selection and physics parameter inference.

NeoWorld aims to balance efficiency and realism by unfolding only the necessary parts of a scene into 3D, enabling interactive world expansion and simple physics-based object manipulation (MPM). Evaluation is performed on the WorldScore and WonderWorld benchmarks.

**Strengths:**

Ambitious problem scope: Tackles interactive, incremental world generation — a compelling and forward-looking problem bridging image-to-3D, scene synthesis, and agentic interaction.

Conceptual novelty: The idea of progressive 2.5D → 3D unfolding is elegant and well-motivated. It provides a practical trade-off between full 3D reconstruction and efficient view synthesis.

System design: The combination of object-centric Gaussian layers, selective 3D conversion, and LLM-based interaction is well engineered and demonstrates clear qualitative improvements over prior 2D/2.5D methods (e.g., WonderWorld, WonderJourney, Matrix-Game2). in particular, having the LLM predict object parameters from an MPM solver, then rendering the result, is elegant and works better than baselines.

Comprehensive qualitative results: The paper covers a wide range of baselines (Wan 2.1-I2V, Matrix-Game2, WonderJourney, WonderWorld) and provides illustrative visual and interactive demos in the supplement.

Human evaluation and benchmarks: Employs WorldScore and multiple human-study-based metrics (3D-Const, SceneQuality, PromptAlign), showing meaningful gains in perceived realism and interactivity.

Clear presentation: The architecture and pipeline are easy to follow, and figures effectively convey the unfolding process.

**Weaknesses:**

1. System-level composition, limited algorithmic novelty.
The method primarily integrates existing models (OneFormer, Marigold, Amodal3R, SD-XL, Gemini LLM) with minimal new learning formulations. The core contributions are system-level engineering rather than new representation learning or theory.

Ablations missing for key design choices e.g. why MPM, why a particular depth estimator was chosen.
No evaluation of alternative inpainting models (e.g., Flux, SD-3.5) or justification for choosing SD-2.0/distilled SD-XL.

Missing ablations for the pose-token alignment module, LLM-based object selection, and codebook design (size, per-scene vs global, dimensionality).

2. Unclear details and justifications.
Line 192: unclear whether the codebook is optimized per scene or shared globally.
Line 280: rationale for model choices (e.g., older SD version, specific normal/depth backbones) is not explained.
LLM choice (Gemini-2.5pro) may give the system an advantage unrelated to algorithmic contribution.

3. Evaluation and benchmarking gaps.

WorldScore and WonderWorld metrics are new and not widely validated; results should be complemented with standard metrics (FID, LPIPS, depth/normal consistency, temporal smoothness, etc.).

Baselines appear visually weaker than their official results; comparisons should use the same scenes, prompts, and camera trajectories for fairness if possible.

4. Quantitative analysis missing.
No per-module timing/memory breakdown
No scaling study of runtime or memory vs. world size or number of unfolded objects.
No quantitative failure attribution across submodules (segmentation, depth, 3D reconstruction, LLM).
Missing physics/animation metrics — dynamics and prompt following are only shown qualitatively.

**Questions:**

1. In the move object examples in the supplemental video, the synthesized results don’t appear to harmonize well with the rest of the scene e.g plausible shadows underneath the boat or chairs. Why is this?

2. The presented samples in Fig. 3 clearly favor this method, but there appears to be a large quality gap between presented results of WonderWorld and what is shown on their project page. Why is this? Could the authors compare on the same scenes that they use in their paper for apples to apples?

3. Does the method work on test images with variation in style, domain, and resolution (e.g., indoor/outdoor, toon vs. photorealistic)?

4. Figure 11: there are several papers that can do inpainting. The authors present a distilled SDXL to do this. How does the distilled model compare with benchmarks in performance and quality?

---

> ### Author Response · Authors · 2025-11-24
> **[Part 1] Official Comment by Authors**
>
> We thank the reviewer for the constructive comments. Please refer to our revised paper for the newly added results and corresponding visualizations.
>
> > Q1. The method primarily integrates existing models (OneFormer, Marigold, Amodal3R, SD-XL, Gemini LLM) with minimal new learning formulations. The core contributions are system-level engineering rather than new representation learning or theory.
>
> (1) *The core contributions in representation learning.*
>
> We would like to clarify that our method does introduce a novel representation learning component, **object-centric Gaussian layers**, specifically tailored for interactive unbounded world generation. This setting imposes several unique requirements on the representation:
> - (i) it must be compact in 3D,
> - (ii) it must handle online, continually arriving data, and
> - (iii) it must remain consistent across multiple scenes.
>
> In contrast, prior object-centric methods (e.g., GaussianGrouping and OmniSeg3DGS) are designed for offline settings: they first segment all 2D images and then train on the full dataset. Existing representation learning approaches like LangSplat and EmerNeRF rely on autoencoders, PCA, or linear mappings for feature compression, which tend to suffer from catastrophic forgetting in online scenarios. To address these challenges, we introduce a globally shared **codebook representation** together with a **cross-scene alignment mechanism**, which enables efficient, object-centric modeling in interactive unbounded world generation while maintaining cross-scene consistency and scalability (see **Table 6**).
>
> (2) *Why system-level composition rather than new learning formulations?*
>
> Our method relies on multiple pretrained models because no single representation currently supports simultaneous accurate 3D reconstruction, high-quality 2D generation, and an object-level interface. Existing unbounded world generation methods that incorporate 3D structure (such as WonderJourney, WonderWorld, and InfiniCube) also necessarily rely on a composition of pretrained models.
>
> To mitigate error accumulation and improve robustness, unlike WonderWorld, we design a **fallback strategy** that prevents the entire system from degrading severely when individual components fail. Also, we initially experimented with end-to-end training to address this, but fine-tuning and co-optimizing the modular design proved computationally expensive. The fallback strategy is thus a practical compromise that balances robustness and efficiency.
>
> > Q2. Ablations missing for key design choices e.g. why MPM, why a particular depth estimator was chosen. No evaluation of alternative inpainting models (e.g., Flux, SD-3.5) or justification for choosing SD-2.0/distilled SD-XL.
>
> (1) *Why MPM?*
>
> We adopt the MPM because it is a well-established framework for simulating a broad range of materials (elastic objects, fluids, sand, snow, and codimensional structures) and has efficient GPU implementations. Importantly, existing works that couple 3D Gaussian Splatting with physics-based simulation also predominantly use MPM, making it a natural, compatible choice. MPM’s particle-based representation aligns well with our Gaussian Splatting framework, allowing us to inject latent physical dynamics into diverse scenarios using a unified particle representation.
>
> (2) *Evaluation of alternative inpainting models.*
>
> In Sec. 4.5 (**Table 5**) and Appendix B, we examine the impact of different depth/normal estimators, inpainting models, image-to-3D models, and LLMs in NeoWorld. The results show that even lighter models deliver acceptable performance, demonstrating the robustness of our approach, while stronger models (e.g., PPD for depth and Marigold for normals) consistently yield better results. Our final selections balance both performance and the efficiency requirements of an interactive system. In addition, we observe that using a stronger inpainting model such as Flux-Fill does not lead to further improvements, primarily because Flux-Fill is trained for object replacement and small-scale inpainting, which does not align well with the large-area completion needed in NeoWorld.
>
> In the "Ablation study of object removal" paragraph in Appendix B, we compare **distilled SD‑XL against LaMa, SD‑2 inpainting, and SD‑XL inpainting**. The distilled SD‑XL model, which is specifically trained for removal, more reliably avoids hallucinating new objects in the erased regions, which is crucial for controllable and coherent scene editing. Moreover, distillation allows inference with fewer sampling steps, further improving the overall efficiency of our method. These ablations jointly motivate our final choice of pretrained components.

---

> ### Author Response · Authors · 2025-11-24
> **[Part 2] Official Comment by Authors**
>
> > Q3. Missing ablations for the pose-token alignment module, LLM-based object selection, and codebook design (size, per-scene vs global, dimensionality).
>
> We refer the reviewer to **Appendix B**, where we conduct the ablation study of object alignment, LLM-based object selection, and codebook design (with respect to one-hot encoding, linear mapping, and autoencoder) and corresponding hyperparameter settings(dimensionality).
>
> In this revision, we additionally include experiments on codebook size and per-scene vs. global codebooks (see **Table 9**). The results show that, as long as the global codebook size exceeds the typical number of objects, the performance is largely stable, and the codebook size has a negligible impact on world storage. In contrast, using per-scene codebooks leads to noticeable performance drops due to feature ambiguity across scenes, which introduces noise and errors in the object segmentation.
>
> > Q4. Unclear details and justifications. Line 192: unclear whether the codebook is optimized per scene or shared globally. Line 280: rationale for model choices (e.g., older SD version, specific normal/depth backbones) is not explained. LLM choice (Gemini-2.5pro) may give the system an advantage unrelated to algorithmic contribution.
>
> In the revision, we have clarified these points:
>
> - **Global codebook.** We explicitly state at Line 192 that the object codebook is shared globally across scenes, rather than optimized per scene. This design is crucial for preserving object-centric accuracy in our interactive, unbounded setting.
> - **Model choice and ablations.** We added a model choice ablation in Sec. 4.5, where we systematically compare different depth/normal estimators, inpainting models, and image‑to‑3D backbones. The results show that lighter models still provide acceptable performance, while stronger models (e.g., PPD for depth and Marigold for normals) yield better quality. These results justify the specific backbones used in the main paper.
> - **LLM choice and its impact.** We use Gemini‑2.5Pro mainly for its stronger reasoning ability and robustness to complex/ambiguous prompts in dynamic interaction. The core quantitative results in Table 1, Table 2, and Table 3, however, are reported without using any LLM; the gains there come directly from our object‑centric scene structure and progressive 3D unfolding. Moreover, thanks to our object-centric representation, the LLM only needs to map user prompts and text-encoded scene information to a limited set of control signals. As shown in Appendix (Table. 14), lighter open‑source models (e.g., Qwen3‑30B‑A3B‑thinking‑2507 and Qwen3‑8B‑thinking) achieve very similar behavior. This indicates that our improvements are not tied to a specific proprietary LLM and that the algorithmic contributions lie in the scene representation and control interface rather than the choice of LLM.
>
> > Q5. Evaluation and benchmarking gaps. WorldScore and WonderWorld metrics are new and not widely validated; results should be complemented with standard metrics (FID, LPIPS, depth/normal consistency, temporal smoothness, etc.).
>
> We appreciate your concern. In the revision (**Table 2**), we report additional metrics to provide a more comprehensive evaluation of NeoWorld. Specifically, we include:
> - **sFID:** measuring the semantic and structural consistency between generated results and real data,
> - **Motion smoothness, subject consistency, and background consistency from VBench:** VBench is a widely used benchmark for video quality evaluation (see Table 2). NeoWorld achieves the lowest sFID and the highest VBench scores among all methods, further demonstrating its advantages in visual fidelity, temporal coherence, and overall scene consistency.
>
> Regarding LPIPS, it requires paired ground-truth images, which are unavailable in our world generation setting. As for depth/normal consistency, the evaluation would depend heavily on specific pretrained depth/normal estimators and is not the main focus of our work.

---

> ### Author Response · Authors · 2025-11-24
> **[Part 3] Official Comment by Authors**
>
> > Q6. Baselines appear visually weaker than their official results; comparisons should use the same scenes, prompts, and camera trajectories for fairness if possible.
>
> We would like to clarify that our evaluation strictly follows the original setup of WonderWorld and WorldScore. The scenes are taken from the WorldScore and WonderWorld datasets, the camera trajectories are identical to those used in WonderWorld, and the prompts are generated using the official WonderWorld code. All models are evaluated with the same prompts and camera trajectories.
>
> The baselines appear visually weaker mainly because we simultaneously showcase both the exploration views and novel **orbiting views** (i.e., 3D rotations around a target object), whereas their official results typically only display the exploration views.
>
> In this revision, we have added a comparison focusing solely on exploration views (see **Fig. 11**). The exploration-view results are consistent with the official baseline visualizations, and WonderWorld and NeoWorld exhibit comparable performance in this setting. We hope this addresses your concerns about the fairness of the comparison.
>
> > Q7. Quantitative analysis is missing. No per-module timing/memory breakdown. No scaling study of runtime or memory vs. world size or number of unfolded objects. No quantitative failure attribution across submodules (segmentation, depth, 3D reconstruction, LLM). Missing physics/animation metrics — dynamics and prompt following are only shown qualitatively.
>
> Thank you for the suggestion. In this revision, we have added a scaling analysis of runtime and memory with respect to both the world size and the number of unfolded objects (**Table 15**), a per-module time breakdown (**Table 18** and Appendix E.4), and failure attribution across submodules (**Table 19 and Appendix F**),  providing a more detailed characterization of NeoWorld’s scalability and efficiency.
>
> For the quantitative evaluation of interactive dynamic world animation, we have already reported results in Table 4 in the original submission, measuring performance in terms of **Prompt Alignment** (a human study assessing text–video similarity) and **VideoCLIP Similarity**, an automated metric computed using VideoCLIP-XL.
>
> > Q8. In the move object examples in the supplemental video, the synthesized results don’t appear to harmonize well with the rest of the scene, e.g, plausible shadows underneath the boat or chairs. Why is this?
>
> The lack of harmonization mainly stems from the fact that we generate dynamics in 3D using MPM-based manipulation. This 3D-centric design has an inherent limitation: it does not explicitly model environment-dependent texture changes such as lighting, shadows, or reflections caused by object–scene interactions.
>
> To further improve realism, one could (i) perform re-shading in 3D space to better account for lighting changes, and/or (ii) integrate 2D video-to-video refinement models on top of our 3D results to enhance visual quality and coherence. In **Fig. 6** and the **supplementary MP4 files**, we present visual enhancement results obtained with V2V models, which effectively improve scene coherence and visual quality, producing natural lighting, shadows, and other appearance effects.
>
>
> > Q9. The presented samples in Fig. 3 clearly favor this method, but there appears to be a large quality gap between presented results of WonderWorld and what is shown on their project page. Why is this? Could the authors compare on the same scenes that they use in their paper for apples to apples?
>
> As mentioned in Q6, the apparent quality gap primarily comes from the difference between exploration views and the novel **orbiting views**.
>
> In Fig. 11, we present comparisons on the same scenes of exploration views, providing an apples-to-apples setting. In Fig. 10, we further showcase the scenes under both exploration and novel views. While WonderWorld and NeoWorld produce comparable quality on exploration views, NeoWorld delivers substantially better results on novel views—precisely where our hierarchical scene representation offers clear advantages.
>
>
> > Q10. Does the method work on test images with variation in style, domain, and resolution (e.g., indoor/outdoor, toon vs. photorealistic)?
>
>
> Yes. In Appendix C, we describe our benchmarks in detail, covering **7 distinct styles and occlusion conditions**, and report NeoWorld’s performance across different scene categories. The results show that NeoWorld generalizes well across these domains.
>
> Regarding resolution, following prior work, all input images are resized to a fixed resolution (e.g., 512×512).

---

> ### Author Response · Authors · 2025-11-24
> **[Part 4] Official Comment by Authors**
>
> > Q11. Figure 11: there are several papers that can do inpainting. The authors present a distilled SDXL to do this. How does the distilled model compare with benchmarks in performance and quality?
>
> In the paragraph "Ablation study of object removal" in Appendix B, we compare our distilled SDXL with other inpainting models(LaMa, SDXL inpainting, SD2 inpainting) in terms of visual quality and unintended object emergence. The results show that all methods achieve comparable visual quality, and the distilled SDXL notably reduces semantic artifacts and unintended new objects, which is crucial for controllable and coherent scene editing.
>
> We therefore choose the distilled SDXL mainly because: (i) it is explicitly trained for removal, making it more reliable at avoiding hallucinated objects in erased regions, and (ii) distillation enables inference with fewer sampling steps, further improving the overall efficiency of our system.

---

### Official Review · Reviewer_VA4X · 2025-11-08

**Soundness:** 2
**Presentation:** 2
**Contribution:** 2
**Rating:** 4
**Confidence:** 3

**Summary:**

The paper proposes a system for interactive 3D world generation from a single image. It introduces an object-centric representation that starts with a layered foreground and background separation and progressively reconstructs objects as 3D Gaussians. The system supports text-driven exploration and manipulation of scene context via LLMs. The paper creates a benchmark dataset based on WonderWorld, WorldScore and WonderJourney, and shows better performance based on human evaluation on CLIP-based metrics.

**Strengths:**

1.Object-centric representation provides greater flexibility in terms of interactive editing and semantic manipulation.
2. The progressive unfolding strategy allows lightweight scene initialization and efficient expansion based on user interaction.

**Weaknesses:**

1. The pipeline depends heavily on pretrained models and heuristics, such as, object alignment, fallback strategy for 3D reconstruction, without any unifying learning based approach.
2. While text-guided object control is well integrated, prior works such as Instruct-NeRF2NeRF, SceneTeller, etc. show similar capability.
3. The paper does not include qualitative video results which makes it difficult to judge the quality of the progressive expansion.
4. There are no ablation experiments or robustness analysis of different pretrained model choices.

**Questions:**

1. How does performance degrade with different model choices, e.g., with lighter models?
2. How robust is the text-based control to ambiguous instructions?

---

> ### Author Response · Authors · 2025-11-24
> **[Part 1] Official Comment by Authors**
>
> We appreciate the reviewer’s insightful comments. Please refer to our revised paper for the newly added results and corresponding visualizations.
>
> > Q1. The pipeline depends heavily on pretrained models and heuristics, without any unifying learning based approach.
>
> The task of unbounded interactive world generation involves several challenging subgoals, including 3D reconstruction, object-level interaction, and camera-controlled scene transitions. However, there does not exist a representation that can simultaneously support all of these capabilities. This mismatch leads to existing systems (e.g., WonderWorld, WonderJourney), including our NeoWorld, necessarily decomposing the problem into multiple subtasks and relying on specialized pretrained models and heuristic designs.
>
> We indeed observed that multi-module pipelines can introduce error accumulation. Initially, we also experimented with an end-to-end training approach to address this issue, but fine-tuning and co-optimizing the modular design proved to be computationally expensive. As a practical compromise, we design a fallback strategy that prevents the system from degrading severely when individual components fail (e.g., segmentation, image-to-3D, or alignment errors). As shown in Table 11 and Fig. 17, this strategy effectively filters out failure cases and is particularly beneficial for maintaining stable performance.
>
> > Q2. While text-guided object control is well integrated, prior works such as Instruct-NeRF2NeRF, SceneTeller, etc. show similar capability.
>
> We would like to clarify the differences from these prior works. Instruct-NeRF2NeRF focuses on appearance editing of existing static or dynamic NeRFs, and does not support an interactive or dynamic object-level interface. SceneTeller, on the other hand, targets single-scene generation rather than unbounded environments.
>
> By comparison, our method enables interactive unbounded world generation, where users can iteratively expand and manipulate the environment, and control object dynamics through natural language commands. This interactive, object-centric interaction in an unbounded setting is not supported by the aforementioned methods.
>
> A side-by-side comparison is summarized below:
>
> | Method              | Scene scale          | Interactivity                  | Object-level control                          | Dynamics                      |
> |---------------------|----------------------|---------------------------------|-----------------------------------------------|--------------------------------|
> | Instruct-NeRF2NeRF | Single scene | Non-interactive edits | Appearance editing of existing objects only   | Static / pre-recorded motion  |
> | SceneTeller        | Single scene         | Non-interactive generation     | Limited object placement                      | Static                         |
> | **Ours**           | Unbounded world      | Interactive, iterative updates | Placement and dynamics |  Controllable dynamics |
>
>
> > Q3. The paper does not include qualitative video results which makes it difficult to judge the quality of the progressive expansion.
>
> We appreciate your concern. Qualitative video results, including visualizations of the interactive world generation and dynamic world animation, have already been provided in the supplementary material. We hope these examples help clearly demonstrate the quality and effectiveness of our progressive expansion process.
>
>
> > Q4. There are no ablation experiments or robustness analysis of different pretrained model choices. How does performance degrade with different model choices, e.g., with lighter models?
>
> In Sec. 4.5 and Appendix B, we investigate the impact of various pretrained components in NeoWorld, including
> - depth estimators, e.g, ZoeDepth and PPD;
> - normal estimators, e.g, DSINE;
> - inpainting models, e.g, LaMa, SD1.5, and Flux-Fill;
> - image-to-3D models, e.g, SplatterImage;
> - and LLMs, e.g, Qwen3-8B, Qwen3-30B-A3B, and Gemini2.5Pro.
>
> As shown in **Tables 5 \& 14** in the revision, we experiment with integrating either lighter or stronger modules. Lighter models achieve satisfactory performance, demonstrating the robustness of our pipeline to backbone choice, while stronger models (e.g., PPD for depth and Marigold for normals) consistently improve result quality. Our final configuration balances performance with the efficiency requirements of an interactive system.
>
> Additionally, in the "Ablation study of object removal" paragraph (**Table 12**), we further include comparisons with LaMa and the non-distilled SDXL model. The distilled SDXL, specifically trained for object removal, more reliably prevents hallucination in erased regions, a critical property for controllable and coherent scene editing. Distillation also reduces the number of required sampling steps, improving inference efficiency. These ablations collectively justify our selection of pretrained components.

---

> ### Author Response · Authors · 2025-11-24
> **[Part 2] Official Comment by Authors**
>
> > Q5.How robust is the text-based control to ambiguous instructions?
>
> We refer the reviewer to Appendix B, “Analysis of LLMs behaviors”, where we specifically analyze how the system handles ambiguous and complex instructions. In this paragraph, we expose the model’s internal reasoning process and evaluate its robustness under under-specified user commands.
>
> The provided example demonstrates that the LLM resolves ambiguity by leveraging spatial context, object salience, and domain priors. As discussed there, the system consistently produces plausible and physically grounded decisions even when the instructions are incomplete or ambiguous, indicating robust text-based control in such scenarios.

---

### Author Response · Authors · 2025-11-22

Dear AC and Reviewers,

Thank you very much for taking the time to review our paper and provide detailed and constructive feedback. We are currently working on adding the corresponding experiments and clarifications to address your comments. However, we still need about 1–2 more days to complete these experiments and carefully verify the conclusions. We hope for your understanding, and we will provide the updated results and a thorough rebuttal as soon as the experiments are finalized.

Best regards,
Authors

---

### Author Response · Authors · 2025-11-24
**General response**

Dear Reviewers,

We sincerely appreciate your constructive comments. In this revised paper, we have added new experiments and incorporated the following modifications:
- **Visual enhancement (Sec. 3.4 & Fig. 6)**: We additionally incorporate state-of-the-art video-to-video (V2V) models as an optional post-processing step to further improve the visual quality and temporal coherence of the generated dynamics (see Fig. 6 and the revised MP4 video files in the supplementary materials).
- **Additional evaluation metrics (Sec 4.1 & Table 2)**: Results with more commonly used metrics, including sFID, and three VBench scores.
- **Analysis on module choices (Sec. 4.5 & Appendix B)**: A comparison of different depth/normal estimators, inpainting models, image-to-3D backbones, removal models, and LLMs.
- **Analysis on the codebook size (Table 9)**: Ablations on codebook size and on per-scene vs. global codebooks.
- **Scalability analysis (Table 15)**: A scaling analysis of runtime and memory with respect to both the world size and the number of unfolded objects.
- **Multi-object manipulation (Table 13 & Fig. 13)**: Quantitative evaluation of our method under multi-object manipulation together with corresponding qualitative visualizations.
- **Multi-view results (Fig. 14)**: Visualizations of dynamic scenes from multiple static cameras and multiple moving cameras.
- **Per-module time breakdown (Appendix E.2 & Table 18)**: A per-module runtime breakdown of our pipeline, providing a more detailed characterization of its overall efficiency.
- **Failure attribution across submodules (Table 19 & Appendix F):** Analysis of the empirical failure frequency of each module and representative failure cases.
- **Additional qualitative results (Fig. 10 & 11):** Interactive world generation results with more comprehensive comparisons with the baselines.


Please do not hesitate to let us know of any additional comments on the paper.

---

### Author Response · Authors · 2025-12-03
**Summary of rebuttal**

Dear AC,

We sincerely thank you for the time and effort you devoted to reviewing our submission. For your convenience, we provide a brief summary of our rebuttal below:

### **Summarization of Technical Novelty**

NeoWorld is the first work to explore object-centric, unbounded 3D world generation, with text-driven user interfaces. This is enabled by our technical innovations:
- New continuous object-centric representation that supports compact, online, and multi-scene–consistent learning in unbounded interactive worlds—via a learned compact object codebook for efficiency, combined with panoptic matching and KNN smoothing for cross-view continuity.
- Progressive 3D unfolding, which adopts a two-stage alignment strategy to keep balance efficiency and immersion.
- LLM-powered text-grounded object-user interface, enabling convenient and fine-grained user–world interaction.

We emphasize that achieving object-centric, continuous, efficient, and immersive world generation is the central goal of our work—far from a trivial combination of existing approaches.


### **Main Concerns Raised by Reviewers and Our Response**
> (i) System-level composition, without unified learning (Reviewer VA4X, UroN)

(1) The core challenge lies in the fact that a unified 3D, interactive, and extensible representation is still underexplored, including in recent systems such as WonderWorld and WonderJourney.

(2) We introduce a carefully designed fallback strategy that effectively mitigates error accumulation arising from module composition.

> (ii) Without new representation learning (Reviewer UroN)

Our method does introduce a novel representation learning component, **object-centric Gaussian layers**, specifically tailored for interactive unbounded world generation, which requires compact, online, and multi-scene-consistent modeling. This representation enables efficient, object-centric modeling while maintaining cross-scene consistency and scalability (see Table 6).

> (iii) Missing ablations of model choice (Reviewer VA4X, UroN)

In the revision, we have added ablations on model choices for the depth/normal estimator, inpainting model, image-to-3D model, object removal model, and LLM (Tables 5, 12, and 14).



> (iv) The synthesized results don’t appear to harmonize well with the rest of the scene. (Reviewer UroN, kLxv)

The harmonization issue arises because dynamics are generated in 3D using MPM-based manipulation. Realism can be further improved by (i) applying 3D re-shading, and/or (ii) using 2D V2V models to enhance visual quality and coherence (see updated **Fig. 6 and supplementary MP4**).

We have also addressed, in detail in the rebuttal, all other concerns raised by the reviewers, including:
- For **Reviewer VA4X**: differences from Instruct-NeRF2NeRF and SceneTeller, qualitative video results, and robustness to ambiguous instructions.
- For **Reviewer UroN**: ablations for pose–token alignment, LLM-based object selection, and codebook design; the rationale for using MPM; clarification of codebook details and design choices; evaluation and benchmark gaps; missing quantitative analysis; comparison fairness; and model generalization.
- For **Reviewer kLxv**: the limitation that object movement is primarily language-driven, and whether reconstruction quality can be further improved.
- For **Reviewer rktf**: support for dynamic objects under a dynamic camera, and performance degradation with respect to the number of objects.

All of these points are systematically discussed and supported with additional experiments and analysis in the rebuttal.

### **Major Modifications in the Revision**
In this revision, our main changes are as follows:
- **Visual enhancement (Sec. 3.4 & Fig. 6)**: We additionally integrate state-of-the-art video-to-video (V2V) models as an optional post-processing step to further improve the visual quality and temporal coherence of the generated dynamics (see Fig. 6 and the revised MP4 videos in the supplementary materials).
- **Additional experimental results**: We include more commonly used metrics (**Table 2**), a detailed analysis of module choices (**Sec. 4.5 & Appendix B**), an analysis of codebook size (**Table 9**), scalability analysis (**Table 15**), multi-object manipulation (**Table 13**), a per-module runtime breakdown (**Table 18**), and failure attribution analysis (**Table 19**).
- **Additional visualizations**: We provide more visual results on multi-object manipulation (**Fig. 13**), multi-view (**Fig. 14**) as well as more comprehensive comparisons (**Figs. 10 & 11**).

For further details, please refer to our point-by-point responses to each reviewer. During the rebuttal, we have added extensive new results. Since the reviewers did not further engage in the discussion phase, we kindly hope you could carefully consider these new results when assessing our work.

Once again, we sincerely thank you for your time and effort in handling our submission.

Best regards,

Authors

---

### Meta-Review · Area_Chair_pnXX · 2025-12-27

**Summary:**

Reviewers acknowledged that the paper addresses an ambitious and relevant problem, namely interactive and unbounded 3D world generation from a single image, and generally found the system design and qualitative demonstrations to be compelling. The proposed progressive 2.5D to 3D unfolding strategy and object centric interaction paradigm were considered well motivated and practically useful.

The suggested decision was primarily informed by recurring concerns regarding the level of algorithmic novelty. Multiple reviewers characterized the work as a system level integration of existing pretrained components rather than introducing fundamentally new learning methods. In addition, the original submission was perceived to lack sufficient ablation studies, quantitative analysis, and justification of key design choices, which weakened confidence in the claimed advantages. Although the rebuttal added substantial experimental results and clarifications, overall reviewer sentiment remained borderline to negative, with no clear consensus toward acceptance.

**Reviewer Concerns:**

**Addressed Concerns**

- Missing ablations on model choices and design decisions (Raised by reviewers VA4X and UroN):
The authors added extensive ablations covering depth and normal estimators, inpainting models, image to 3D backbones, language models, object removal strategies, codebook design, and alignment modules.
- Lack of scalability, efficiency, and robustness analysis (Raised by reviewers UroN and rktf):
The revision includes runtime and memory scaling analyses, per module timing breakdowns, and performance evaluation with increasing numbers of objects.
- Evaluation relying primarily on newly proposed or non standard metrics (Raised by reviewer UroN):
Additional standard metrics, including sFID and VBench, were added to complement WorldScore and human evaluation results.
- Limited evidence for multi object manipulation and support for dynamic cameras (Raised by reviewer rktf):
The authors added both quantitative and qualitative results demonstrating multi object manipulation and object motion under moving camera trajectories.
- Robustness of language based control under ambiguous instructions (Raised by reviewer VA4X):
Additional analysis was provided on language model behavior under ambiguous or underspecified prompts.

**Outstanding Concerns**
- Limited algorithmic novelty beyond system integration (Raised by reviewers VA4X and UroN):
Although the authors clarified the object centric representation and codebook design, the contribution may still be perceived as primarily engineering driven rather than introducing new learning formulations.
- Visual realism and physical plausibility of generated dynamics (Raised by reviewers UroN and kLxv):
Issues related to lighting, shadows, and scene harmonization remain, and improvements rely in part on optional post processing rather than being fully addressed within the core model.
- Limited quantitative evaluation of motion realism and physical behavior (Raised by reviewer kLxv):
Motion quality and physical realism are still evaluated largely qualitatively, with few task specific quantitative metrics.

**Reviewer Scores:**

- Reviewer VA4X (initial score: 4):
While the rebuttal addressed several technical and experimental concerns, the primary concern regarding limited algorithmic novelty would likely remain. The score would most likely remain at 4. If the reviewer were to revise the score, it would most plausibly increase to 6 rather than to an intermediate value.
- Reviewer UroN (initial score: 4):
Although extensive ablations and clarifications were added in the rebuttal, the core concern that the work represents system level integration rather than introducing new learning methods would likely persist. The score would most likely remain at 4.
- Reviewer kLxv (initial score: 6):
The rebuttal does not substantially alter the reviewer’s assessment of realism and reconstruction quality. The score would likely remain unchanged at 6.
- Reviewer rktf (initial score: 4):
Additional experiments on multi object manipulation and dynamic cameras address some issues but do not fundamentally change the overall assessment. The score would most likely remain at 4. If revised, it would most plausibly increase directly to 6.

---

### Decision · Program_Chairs · 2026-01-26

Reject